# A bioactive peptide amidating enzyme is required for ciliogenesis

Dhivya Kumar[1][†], Daniela Strenkert[2], Ramila S Patel-King[1], Michael T Leonard[2], Sabeeha S Merchant[2,3], Richard E Mains[4], Stephen M King[1]*, Betty A Eipper[1,4]*

[1]Department of Molecular Biology and Biophysics, University of Connecticut Health Center, Farmington, United States; [2]Department of Chemistry and Biochemistry, University of California Los Angeles, Los Angeles, United States; [3]Institute for Genomics and Proteomics, University of California, Los Angeles, Los Angeles, United States; [4]Department of Neuroscience, University of Connecticut Health Center, Farmington, United States

*For correspondence: eipper@ uchc.edu (BAE); king@uchc.edu (SMK)

Present address: [†]Department of Biochemistry and Biophysics, University of California San Francisco, San Francisco, United States

Competing interests: The authors declare that no competing interests exist.

**Abstract** The pathways controlling cilium biogenesis in different cell types have not been fully elucidated. We recently identified peptidylglycine α-amidating monooxygenase (PAM), an enzyme required for generating amidated bioactive signaling peptides, in *Chlamydomonas* and mammalian cilia. Here, we show that PAM is required for the normal assembly of motile and primary cilia in *Chlamydomonas*, planaria and mice. *Chlamydomonas* PAM knockdown lines failed to assemble cilia beyond the transition zone, had abnormal Golgi architecture and altered levels of cilia assembly components. Decreased PAM gene expression reduced motile ciliary density on the ventral surface of planaria and resulted in the appearance of cytosolic axonemes lacking a ciliary membrane. The architecture of primary cilia on neuroepithelial cells in *Pam*[-/-] mouse embryos was also aberrant. Our data suggest that PAM activity and alterations in post-Golgi trafficking contribute to the observed ciliogenesis defects and provide an unanticipated, highly conserved link between PAM, amidation and ciliary assembly.

## Introduction

Cilia are ancient microtubule-based organelles derived from the basal body, and were present in the last common ancestor of eukaryotes (*Carvalho-Santos et al., 2011*). The signaling potential of primary cilia and the added ability of motile cilia to generate propulsive force established them as key organelles required for the development and homeostasis of diverse cell types and organisms. The protein and lipid complements of the cilium are distinct from the rest of the cell; more than 700 proteins and specific lipids such as sterols are enriched in this intricate structure. Thus, the biogenesis and maintenance of these complex organelles is a tightly regulated cell type specific event (*Hsiao et al., 2012*; *Nachury et al., 2010*; *Lechtreck, 2015*; *Taschner and Lorentzen, 2016*). Disruption of ciliogenesis or ciliary function leads to ciliopathies, a group of multisystemic diseases that have partially overlapping, often severe phenotypes, highlighting the importance of understanding these processes in different cells (*Brown and Witman, 2014*; *Fliegauf et al., 2007*; *Waters and Beales, 2011*).

The roles of microtubule motors and intraflagellar transport (IFT) proteins in trafficking the cargo proteins required to build and maintain ciliary architecture have been well described. The growing cilium also relies on the delivery of membrane vesicles and proteins derived from the Golgi, and disruption of the secretory pathway by brefeldin A inhibits ciliogenesis (*Dentler, 2013*; *Haller and Fabry, 1998*). Additionally, proteins and complexes involved in polarized vesicular trafficking such as the clathrin adaptor protein-1 complex (AP1), Rabs and Arf/Arl proteins, the exocyst and BBS

**eLife digest** Animal cells produce many small proteins known as peptides that help cells to communicate with each other. To become active, many of these peptides need to be chemically modified. PAM is the only enzyme that carries out a type of peptide modification called amidation and it is essential for animals to grow and survive.

PAM was originally thought to have evolved in the nervous system of animals, but recent studies have found that it is also present in green algae and in hair-like projections known as cilia, which are found on the surface of most animal cells. There are two types of cilia: motile cilia beat rhythmically and are responsible for moving cells and fluids, while non-motile cilia sense the external environment and serve as signaling hubs. These observations suggest that PAM may have other roles in cells in addition to activating peptides.

Kumar et al. now set out to investigate the role of PAM in green algae, flatworms and mice. The results show that PAM is important for the formation of both motile and non-motile cilia. Reducing the levels of PAM in the algae and flatworms resulted in short stubs forming on the surface of cells instead of motile cilia. This reduced the ability of flatworms to glide around their environment. Furthermore, the cells of mutant mice lacking PAM produced non-motile cilia that were much shorter than those produced by normal mouse cells.

Further experiments suggest that PAM may be involved in the transport of certain proteins to the sites where new cilia will form. The findings of Kumar et al. reveal a link between the PAM enzyme, amidation and the assembly of cilia. The next step will be to identify the molecules that are modified by PAM and work out exactly how amidated products might regulate the formation of cilia.

complexes are critical for cilium biogenesis (*Hsiao et al., 2012*; *Nachury et al., 2007*; *Zuo et al., 2009*). The signaling processes coordinating these pathways and regulating ciliogenesis are poorly understood.

Peptidylglycine α-amidating monooxygenase (PAM), a secretory pathway-localized enzyme, catalyzes one of the final steps in the biosynthesis of many signaling peptides (*Kumar et al., 2016a*). PAM-catalyzed C-terminal amidation confers bioactivity to secreted peptides such as oxytocin, vasopressin and neuropeptide Y. The two enzymatic domains of PAM act sequentially on glycine-extended peptide precursors. Peptidylglycine α-hydroxylating monooxygenase (PHM) catalyzes the copper and ascorbate dependent hydroxylation of the α-carbon of the terminal glycine, and peptidyl-α-hydroxyglycine α-amidating lyase (PAL) cleaves the N-C bond, generating glyoxylate and the α-amidated peptide. PAM is a type I integral membrane protein containing a cytosolic domain that is not essential for catalytic activity, but is necessary for routing the enzyme through the secretory and endocytic pathways (*Milgram et al., 1993*).

Based on a phylogenetic study identifying PAM-like genes in green algal genomes, we recently demonstrated the presence of active enzyme in the unicellular eukaryote, *Chlamydomonas reinhardtii* (*Attenborough et al., 2012*; *Kumar et al., 2016b*). Despite the evolutionary distance between green algae and mammals, the biochemical properties of *C. reinhardtii* PAM (CrPAM) are remarkably similar to those of rat PAM. In both species, the full-length enzyme is membrane tethered, with its two catalytic domains, PHM and PAL, residing in the secretory pathway lumen. We also demonstrated that the catalytic domains of CrPAM can be separated from its transmembrane and cytosolic domains, leading to the generation of soluble bifunctional enzyme that can be secreted from cells (*Kumar et al., 2016b*).

The striking evolutionary co-occurrence of organisms containing PAM-like genes and cilia prompted us to explore PAM localization in *C. reinhardtii*. Using an antibody recognizing the cytosolic domain of CrPAM, we found that this enzyme is present in the Golgi and cilia (*aka* flagella). PAM was also observed in motile and primary cilia of mammalian cells (tracheal epithelial cells, fibroblasts, spermatozoa) (*Kumar et al., 2016b*). Furthermore, in *C. reinhardtii* cilia, PAM activity displayed an unexpected, strong biochemical association with the axonemal superstructure (*Kumar et al., 2016b*). Together, these observations in multiple cell types suggested that PAM has a novel and highly conserved signaling or sensory function in eukaryotic cilia. Here we demonstrate

that PAM plays a key conserved role during the early steps of ciliogenesis, revealing a novel link between amidation and cilium assembly in multiple cell types.

## Results

### Knockdown of PAM expression disrupts ciliogenesis in *C. reinhardtii*

To explore the function of PAM in *C. reinhardtii*, we designed an artificial microRNA (amiRNA2) targeted to the 5' region of the CrPAM gene (*Figure 1A*); expression was under control of the strong, constitutive promoter, HSP70A-RBSC2 (*Molnar et al., 2009*; *Schroda et al., 2000*). CrPAM protein expression was assessed by western blot analysis of whole cell lysates; seven transformants and two empty vector control transformants were chosen for further phenotypic analysis. Phase-contrast microscopy revealed that all seven knockdown strains were immotile and lacked cilia. Two strains (PAM-amiRNA2 # 3 and # 8) were selected for detailed analysis. Lysates of both strains contained reduced amounts of PAM protein (*Figure 1B*). Enzyme assays showed reduced levels of PHM and PAL activity in both strains (*Figure 1C*). Compared to empty vector transformed control strains, activity was reduced to about 30% and 10% in PAM-amiRNA2 #3 and #8, respectively.

We next used immunostaining for CrPAM and acetylated tubulin to compare PAM-amiRNA and empty vector *C. reinhardtii* cells. Images procured under similar exposure settings confirmed reduction of CrPAM levels in PAM-amiRNA strain #8 when compared to the empty vector control strain (*Figure 1D*). As reported previously (*Kumar et al., 2016b*), most of the PAM protein localized to the Golgi region (*Figure 1D*), while a small fraction (7% of total PAM activity; *Figure 1—figure supplement 1*) was present along the length of the cilia (inset in *Figure 1D*) in the empty vector controls. Most strikingly, staining for acetylated tubulin confirmed the absence of cilia in both knockdown lines. Although cilia were robustly stained in control cells, only cell body microtubules were visible in the PAM-amiRNA cells (*Figure 1D*). To explore the possibility of the formation of short ciliary stubs in the PAM-amiRNA mutants, we utilized scanning electron microscopy. Most control cells had two cilia that were each ~10 μm in length. In contrast, cilia were never observed on cells of either knockdown strain; only short ciliary stubs were visible (inset in *Figure 1E*). PAM-amiRNA cells otherwise appeared morphologically normal in size and shape (*Figure 1E*).

Both knockdown strains grew at the same rate as controls in both acetate-containing and minimal media (*Figure 1—figure supplement 2A*). Thus, growth under both photoautotrophic ($CO_2$ as the sole carbon source assimilated by photosynthesis) and photoheterotrophic (using acetate as a carbon source) conditions was unaltered. This clearly indicates that the PAM-amiRNA cells are generally healthy and do not exhibit major metabolic defects in chloroplast or mitochondrial function. The period of the contractile vacuole cycle, which relies on vesicular trafficking, was indistinguishable in knockdown and control strains grown under standard culture conditions (*Figure 1—figure supplement 2B and C*).

To confirm the specificity of the knockdown, we used another amiRNA (amiRNA1) targeting a different region of the PAM gene (*Figure 1A*), and found that this also lead to reduction of PAM activity and defective ciliogenesis (*Figure 1—figure supplement 3* and *Table 1*). Thus, the observed phenotypes are not the result of off-target effects from any given amiRNA. As the amiRNA1 strains appeared somewhat heterogeneous, presumably due to the transgene silencing that often occurs in *C. reinhardtii* (*Yamasaki et al., 2008*), all subsequent experiments were performed with amiRNA2 strains.

### PAM enzymatic activity is important for ciliogenesis

These results were consistent with a key role for PAM in regulating ciliogenesis in *C. reinhardtii* cells. The cytosolic domain of mammalian PAM is not required for enzyme activity, but controls its endocytic trafficking, and acts as a hub for various cytosolic protein interactions (*Milgram et al., 1993*; *Prigge et al., 1997*). To determine whether this PAM domain was required, we procured an insertional mutant from the indexed *C. reinhardtii* mutant library (*Li et al., 2016a*) in which the CrPAM cytosolic domain was disrupted. The *C. reinhardtii* insertional mutant obtained (CrPAM-ΔCD) encoded a protein lacking the final 73 residues of the 101-residue cytosolic domain of CrPAM; the first 28 residues of the cytosolic domain were followed by an 18-amino acid sequence derived from the insertion cassette (*Figure 2A*). Lysates prepared from the CrPAM-ΔCD mutant strain exhibited

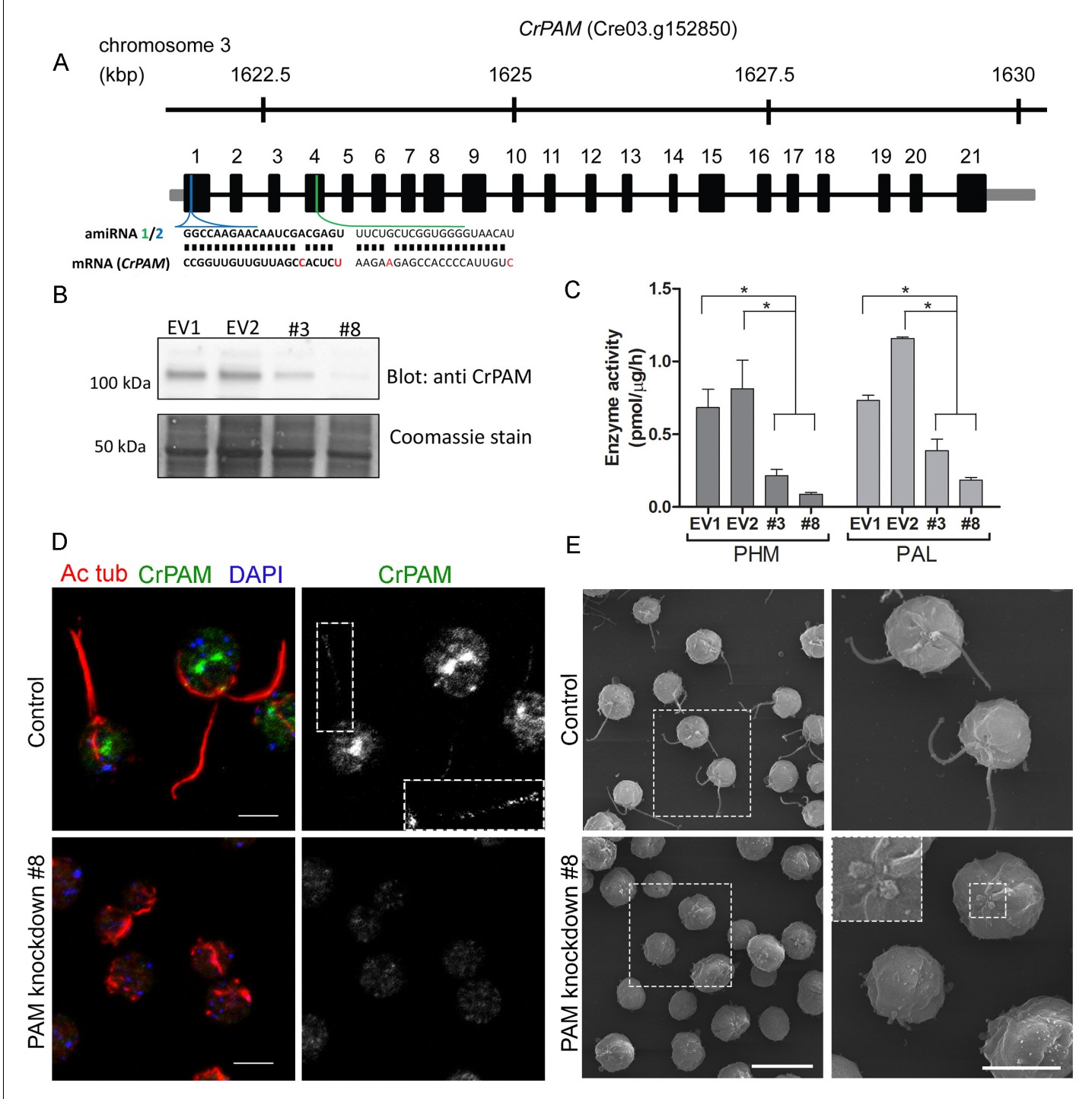

**Figure 1.** Knockdown of PAM disrupts ciliogenesis in *C. reinhardtii*. (A) Schematic showing CrPAM (Cre03.g152850) gene structure and amiRNA target sequences near the 5' end of the CrPAM mRNA (amiRNA1, green; amiRNA2, blue). 5'- and 3'-untranslated regions (gray bars), exons (numbered rectangles) and introns (black lines) drawn to scale. (B) Cell lysates prepared from control strains transformed with empty vector (EV1 and EV2) and knockdown strains transformed with the CrPAM amiRNA2 construct (#3 and #8) were subjected to western blot analysis using an antibody to the C-terminal domain of CrPAM. Top panel shows reduction of CrPAM band intensity (110 kDa) in the two knockdown strains. Coomassie staining of the membrane (bottom panel) shows equal loading. (C) PHM and PAL specific activities in cell lysates from triplicate assays (mean ± SD). Asterisks indicate p<0.05 in a one-way Anova. (D) Immunofluorescence images of *C. reinhardtii* control and PAM amiRNA2 #8 cells stained with antibodies to acetylated tubulin (red) and CrPAM (green) acquired at equal exposure. Right panels show CrPAM staining in the cilium (inset) and Golgi, which is lost in knockdown cells. Acetylated tubulin staining shows loss of cilia; cortical microtubules are still visible in knockdown cells. Scale bar, 5 µm. (E) Scanning
*Figure 1 continued on next page*

*Figure 1 continued*

electron micrographs of control (top panels) and PAM amiRNA2 #8 cells (bottom panels) at low (left panels, scale bar, 10 μm) and high (right panels, scale bar, 5 μm) magnification.

The following figure supplements are available for figure 1:

**Figure supplement 1.** Distribution of PHM activity in cilia and cell bodies of *C. reinhardtii*.

**Figure supplement 2.** Knockdown of PAM does not affect cell growth or the contractile vacuole cycle.

**Figure supplement 3.** Knockdown of *PAM* expression by two different amiRNAs leads to ciliogenesis defects.

significantly more PHM and PAL enzyme activity than the control CC4533 strain (*Figure 2B*). As expected, antibody specific for the cytosolic domain of CrPAM failed to detect any cross-reactive protein in CrPAM-*Δ*CD lysates.

To aid further analysis of this strain, we generated a new antibody to the luminal enzymatic domains of CrPAM (*Figure 2—figure supplement 1*). This affinity-purified antibody detects a band on immunoblots that comigrates with that detected by the CrPAM-CD antibody (*Figure 1—figure supplement 3A*); the intensity of this band is greatly diminished in the knockdown strains compared to controls (*Figure 1—figure supplement 3A*) demonstrating that it indeed represents PAM. Furthermore, the luminal antibody band intensity increased in the PAM knock down strains, consistent with the increase in PAM enzymatic activity (*Figure 2B and C*), and showed an immunofluorescence staining pattern similar to that of the CrPAM-CD antibody (*Figure 1D* and *Figure 2—figure supplement 2*).

Antibody against the luminal domain detected PAM in the perinuclear region of the PAM-*Δ*CD strain while the cytosolic domain antibody did not (*Figure 2—figure supplement 2*). In agreement with the activity measurements, levels of CrPAM-*Δ*CD protein exceeded levels of CrPAM protein, perhaps reflecting its altered trafficking and turnover (*Figure 2B,C*). The CrPAM-*Δ*CD cells were motile, with cilia that were similar in length to control cells (9.4 ± 0.2 μm for CC4533 and 9.5 ± 0.2 μm for CrPAM-*Δ*CD cells; n > 80 for each strain) (*Figure 2D*). Transmission electron microscopy revealed no apparent ultrastructural defects in the cilia of CrPAM-*Δ*CD cells (*Figure 2E*).

The fact that CrPAM does not require an intact C-terminus in order to support ciliogenesis suggests that the catalytic cores may provide the essential ciliogenic factor. To further examine this hypothesis, we tested whether *C. reinhardtii* PHM activity was inhibited by 4-phenyl-3-butenoic acid (PBA), a mechanism-based PHM inhibitor that has been used in vitro and in vivo (*Bradbury et al., 1990*; *Driscoll et al., 2000*; *Katopodis and May, 1990*). PBA did not affect the growth properties of *C. reinhardtii* (*Figure 2F*), but did inhibit both mammalian and *C. reinhardtii* PHM activities (*Figure 2G and H*). When *C. reinhardtii* cells were deflagellated and allowed to regenerate cilia in the presence of PBA, ciliary regrowth was delayed, suggesting that PAM enzymatic activity is important for ciliogenesis (*Figure 2I*). To further confirm this result, we deflagellated cells treated with neocuproine, a highly selective copper chelator (*Mendelsohn et al., 2006*) as the PHM-mediated

**Table 1.** Quantification of ciliated cells in amiRNA knockdown strains. The % ciliated cells in asynchronous cultures was determined by microscopic examination following formaldehyde fixation (n > 100 for all samples). All amiRNA strains had reduced numbers of ciliated cells.

| Strain | % Ciliated cells (n > 100) |
|---|---|
| EV3 | 58% |
| amiRNA1 #5 | 24% |
| amiRNA1 #6 | 8.7% |
| amiRNA2 #8 | 0% |

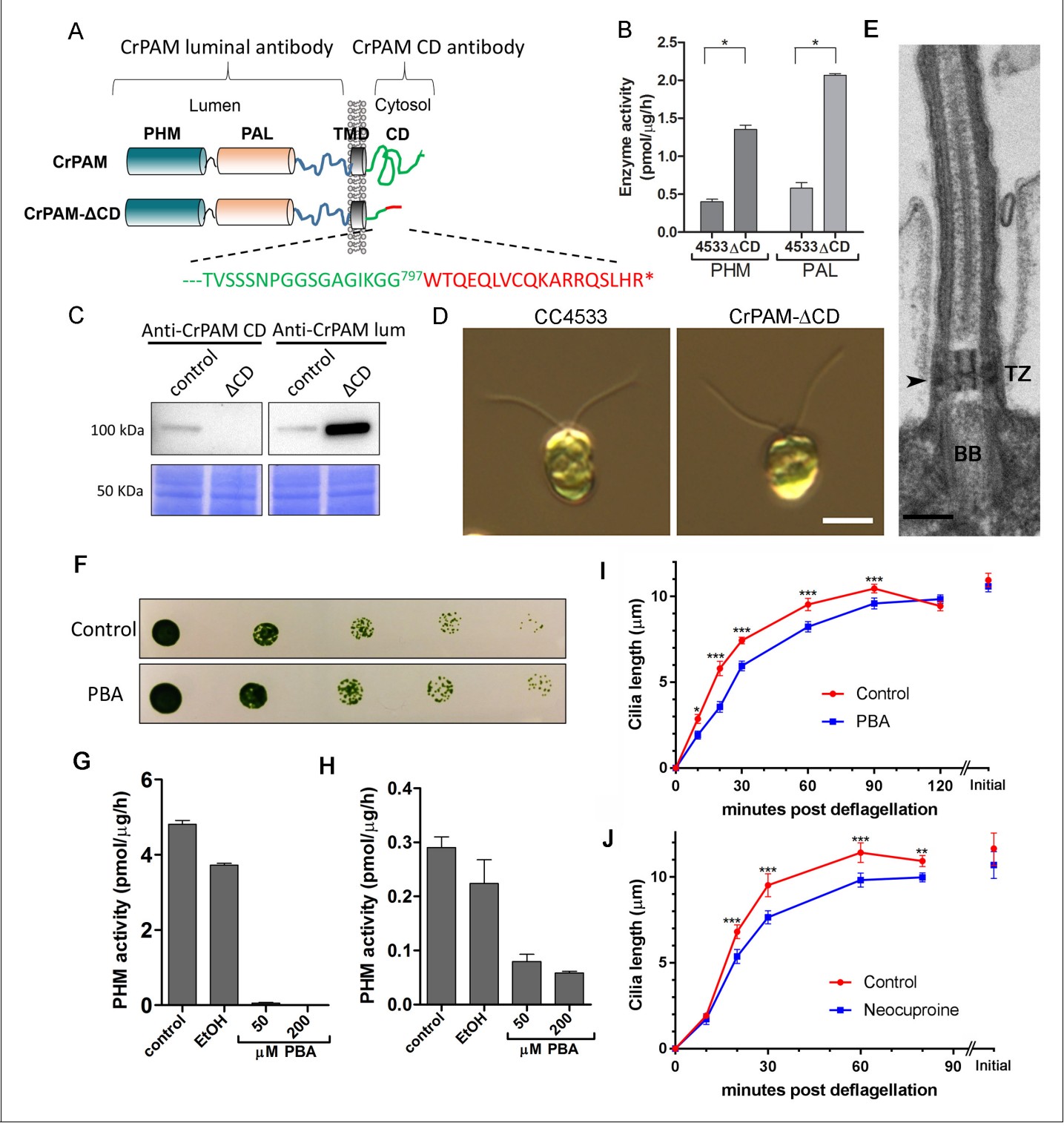

**Figure 2.** *C. reinhardtii* strain lacking the C-terminal domain of PAM assembles cilia. (**A**) Schematic showing disruption of the cytosolic domain of CrPAM in the CrPAM-ΔCD strain. Residues corresponding to the C-terminus of CrPAM (green) and insertion cassette (red) are shown. (**B**) PHM and PAL specific activities in control (CC-4533) and CrPAM-ΔCD strain lysates (mean ± SD). Both PHM and PAL activity increased in CrPAM-ΔCD cells (p<0.05 using unpaired t-test). (**C**) Western blot of control and CrPAM-ΔCD strain lysates with C-terminal domain and luminal antibodies. As predicted by its sequence, CrPAM-ΔCD could be detected by the PAM luminal domain antibody, but not by the PAM cytosolic domain antibody. (**D**) Differential interference contrast images of control and CrPAM-ΔCD cells showing the presence of cilia in both (Scale bar, 5 μm). (**E**) Transmission electron micrograph showing normal ciliary structure in CrPAM-ΔCD strain. Basal body (BB), transition zone (TZ) and Y-linkers (arrowhead) appear normal (Scale

*Figure 2 continued on next page*

*Figure 2 continued*

bar, 200 nm). (F) Growth of 5-fold serially diluted CC-124 wildtype *C. reinhardtii* on TAP plates containing 0 (control) or 200 µM phenylbutenoic acid (PBA) 6 days post-innoculation. (G and H) Cell lysates were prepared from mouse corticotropes expressing rat PAM (G) and from *C. reinhardtii* CC-124 cells (H); PHM assays were carried out after addition of the indicated amount of PBA or vehicle (ethanol, EtOH). PHM specific activity was greatly reduced in both lysates in the presence of PBA. (I and J) Wildtype *C. reinhardtii* CC-124 cells were preincubated for 3 hr with PBA or neocuproine, deciliated by pH shock, which releases cilia immediately distal to the transition zone, and allowed to regrow cilia in the presence of 200 µM PBA (I) or 10 µM neocuproine (J). Error bars represent 95% confidence intervals; 80–100 measurements were made for each point (***p<0.001; **p<0.01; *p<0.05). Differences between control and experimental plots are significant (unweighted means analysis; p<0.0001; variance = 1.94% for PBA and 2.07% for neocuproine). Graphs are from one of two experiments which yielded similar results.

The following figure supplements are available for figure 2:

**Figure supplement 1.** Generation of antibody to the luminal domains of CrPAM.

**Figure supplement 2.** PAM is present in the perinuclear region of PAM-ΔCD cells.

hydroxylation reaction absolutely requires this metal. Neocuproine treatment delayed reformation of ~9.5 µm cilia by 20–30 mins (*Figure 2J*). Together, analysis of the PAM-ΔCD strain and the enzyme inhibitor studies support a role for PAM activity in ciliogenesis.

## *C. reinhardtii* PAM knockdown cells fail to assemble cilia beyond the transition zone

We next assessed the ultrastructural defects resulting from PAM deficiency in *C. reinhardtii* cells. Compared to the normal ciliary morphology observed in longitudinal sections of control cells (*Figure 3A*), we observed only short ciliary stubs in PAM knockdown cells. The basal bodies appeared normal; however, we were unable to observe normal axonemal structures extending beyond the transition zone in these cells (*Figure 3B–E*). Most of the short ciliary stubs seen in PAM knockdown cells contained accumulations of electron dense material; some also had randomly oriented fragments of microtubules (arrows in *Figure 3B and C*). In both longitudinal and cross-sections, most features of the transition zone looked quite similar in PAM-deficient and control cells. Strikingly though, a deficit was apparent in the Y linkers of PAM-deficient cells. Wedge-shaped Y linkers, which connect the microtubules to the membrane and regulate protein entry into the cilium (arrowheads in *Figure 3A*), were readily apparent in longitudinal sections through the transition zone of control cilia. However, we were unable to find Y linkers in the transition zones of PAM knockdown cells (*Figure 3B–D*). Overall, the phenotypes observed in the PAM knockdown cells closely mirrored those previously reported for several IFT and transition zone mutants (*Craige et al., 2010*; *Hou et al., 2007*; *Pazour et al., 2000*).

## PAM is necessary for ciliary motility in metazoans

To test whether this unexpected role for PAM in ciliogenesis was conserved in metazoans, we utilized RNAi-mediated gene knockdown in the planarian *Schmidtea mediterranea*, which has a ciliated ventral epithelium used for gliding locomotion. The *S. mediterranea* genome encodes a bifunctional PAM protein with a transmembrane domain and cytosolic domain, a soluble PHM protein and a soluble PAL protein (*Figure 4A*). To assess the role of bifunctional PAM and soluble PHM, we designed RNAi vectors targeted to each gene (*Figure 4A*); soluble PAL, which would presumably function downstream of soluble PHM, was not targeted. Groups of planarians were fed with bacteria containing either the empty vector control plasmid (L4440), the *Smed-phm(RNAi)*, the *Smed-pam(RNAi)* or a mixture of bacterial cells expressing both the *Smed-phm* and *Smed-pam* RNAi vectors for three weeks. RNA abundance was monitored using RT-PCR. Reductions in PAM and/or PHM mRNA were observed as expected in animals fed with *Smed-phm(RNAi)* or *Smed-pam(RNAi)* or a mixture of both; actin and outer arm dynein IC2 mRNA levels were unaltered (*Figure 4B*). To monitor the success of our knockdown strategy, PHM and PAL enzyme activities were measured in lysates. Both PHM and PAL activities were readily detected in L4440 lysates (*Figure 4C*). PHM activity was reduced to ~40% in *Smed-pam(RNAi)* and *Smed-phm(RNAi)* animals (*Figure 4C*). Knockdown of both *Smed-pam* and *Smed-phm* reduced PHM specific activity to less than 10% of control. As

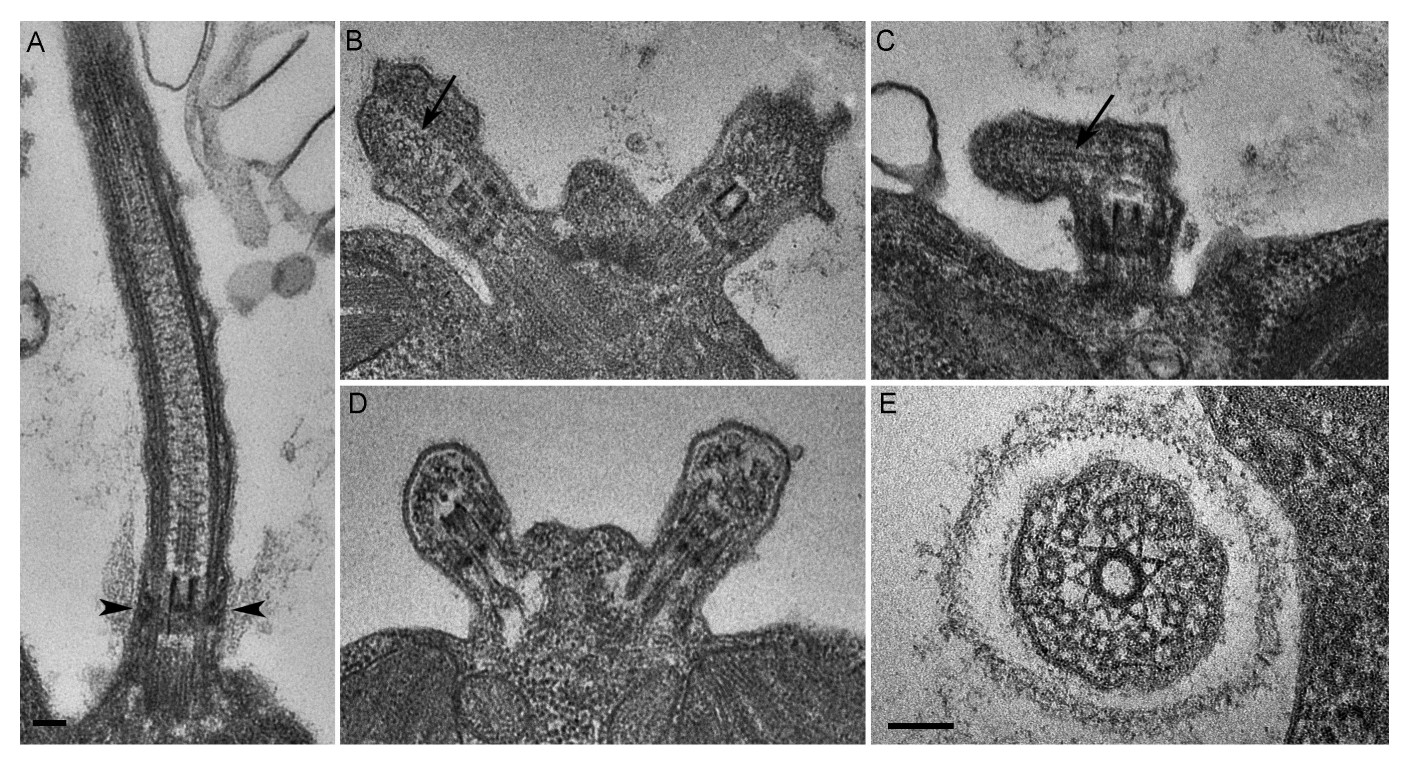

**Figure 3.** *C. reinhardtii* PAM knockdown cells fail to assemble cilia beyond the transition zone. (**A**) Transmission electron micrograph of control cilium showing an ultrastructurally normal basal body, transition zone and axoneme. Wedge shaped Y linkers connecting the microtubules to the membrane in the transition zone are visible (arrowheads). (**B–D**) Examples of ciliary stubs seen in PAM knockdown cells. Longitudinal sections through the anterior portion of the cell show an ultrastructurally normal basal body, although Y linkers appeared to be disrupted. Electron dense material accumulated in the stubs (**B**, **C** and **D**) and microtubule fragments (arrows in B and C) were randomly oriented. (**E**) Cross-section of a transition zone from PAM amiRNA cell showed stellate centrin fibers and outer doublet microtubules surrounded by a ciliary membrane. Electron dense material fills the space between the microtubules and the membrane (Scale bar, 100 nm).

expected, PAL activity was reduced in lysates prepared from *Smed-pam* and the double knockdown animals, but not in *Smed-phm(RNAi)* planaria. Our results also show that most of the PAL enzyme activity was derived from the bifunctional enzyme (*Figure 4C*).

Since cilia are required for gliding locomotion in *S. mediterranea*, we assessed motility using live video microscopy (*Video 1* and *Figure 4D*). Compared to the controls, a greater percentage of the *Smed-phm(RNAi)* and *Smed-phm+pam(RNAi)* animals were immotile (*Table 2*). Interestingly, gliding velocity for *Smed-pam(RNAi)* planaria was comparable to L4440 controls (≈0.7 mm/s), while gliding velocities for *Smed-phm(RNAi)* and the double knockdown animals were lower (≈0.4 mm/s) (*Figure 4E* and *Table 2*). Planarians completely devoid of cilia move at velocities similar to those of the *Smed-phm(RNAi)* and *Smed-phm+pam(RNAi)* animals by using muscular contractions to propel their bodies forward (*Rompolas et al., 2010*). These results suggested that PHM activity was necessary for normal motility behavior in planarians.

To explore the defects in gliding behavior in RNAi-fed planarians, we used high frame-rate video microscopy to analyze ciliary beat frequency (CBF). As observed previously, cilia in L4440 control animals had a CBF of ≈ 26 Hz (*Rompolas et al., 2010*). In PHM and double knockdown animals, CBF was decreased to ≈ 18 Hz and ≈ 17 Hz, respectively. The CBF in the *Smed-pam(RNAi)* knockdown animals was ≈ 30 Hz, consistent with the normal gliding velocity measured for this experimental group (*Figure 4E* and *Table 2*). The waveform of cilia in the double knockdown animals appeared to be dramatically different from control animals, with restricted motion compared to the highly coordinated smooth waveform of control cilia (*Video 2*).

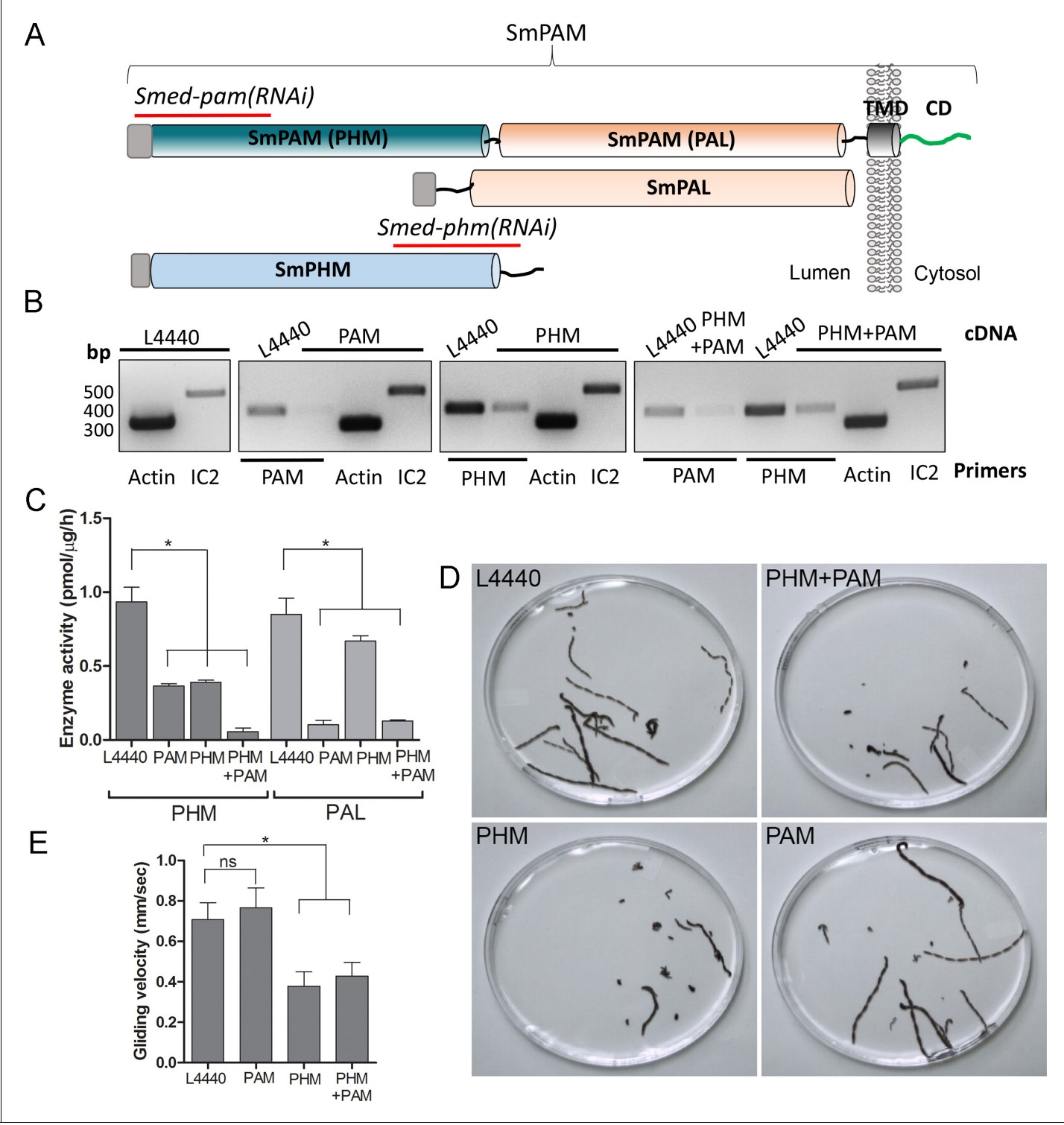

**Figure 4.** Knockdown of PHM gene in planaria causes motility defects. (**A**) Schematic of *S. mediterrania* bifunctional membrane PAM, monofunctional PAL and monofunctional soluble PHM showing the regions used to generate RNAi constructs (red lines). Signal sequences, grey; transmembrane domain in SmPAM, black; cytosolic domain, CD, green. (**B**) RT-PCR analysis of mRNA from flatworms fed with the L4440 (control), PAM, PHM or PHM +PAM dsRNA. The cDNA source is shown above the image, and the primer pairs used for RT-PCR are listed below the image. Actin and dynein IC2 primers were used as controls. (**C**) PHM and PAL specific activities in planarian lysates plotted as mean ± SD from triplicate assays (asterisks indicate p<0.05 in a one-way Anova). (**D**) Overlays of frames from deconvolved videos of control and RNAi planaria showing the tracks taken over 60 s. (**E**) PHM

*Figure 4 continued on next page*

*Figure 4 continued*

and PHM+PAM RNAi animals displayed reduced gliding velocities plotted as mean ± SEM compared to L4440 and PAM RNAi animals (asterisks indicate p<0.05 in an unpaired t test).

## PAM is necessary for ciliogenesis in metazoans

We next assessed ciliary morphology in the groups of planarians fed with bacteria expressing dsRNA. Scanning electron microscopy revealed dense, tightly packed cilia covering the entire ventral surface of L4440 control animals (*Figure 5A*). In stark contrast, knockdown of both PHM and PAM resulted in a severe reduction in ciliary density; planarian cilia undergo constant remodeling and ciliary loss is due to the failure of this process (*Rompolas et al., 2010*) (*Figure 5A*). Flatworms receiving only *Smed-phm(RNAi)* or *Smed-pam(RNAi)* displayed intermediate phenotypes. Ciliary length was very heterogeneous in the RNAi-fed animals - short cilia were interspersed with cilia of normal length, presumably due to variations in the ciliary remodeling process (*Figure 5A*). As loss of PHM activity alone resulted in a ciliogenesis defect in planaria, these results further support a role for amidating activity in this process.

Transmission electron microscopy also revealed pleomorphic defects resulting from the loss of PHM or both PHM and PAM in RNAi-fed planarians. In cross-sections, cilia with a normal 9 + 2 axonemal structure, containing outer and inner dynein arms, were observed in all samples (*Figure 5B–D*). Occasionally, in the *Smed-phm(RNAi)* and *Smed-phm+pam(RNAi)* samples, the doublet microtubule organization was altered; singlet microtubules were sometimes observed along with doublet microtubules and in other cases the number of microtubules was aberrant (*Figure 5E,F*). These aberrant axonemal structures are likely a secondary feature of defective remodeling; they may represent cilia in the process of failing.

In *Smed*-L4440 control samples, basal bodies docked at the plasma membrane and cilia extended outward (*Figure 5G*). Although similar structures were present in the *Smed-phm+pam(RNAi)* sections, some docked basal bodies had no cilia emanating from them (arrowhead, *Figure 5H* and *Table 3*); there was an approximately 30% reduction in the density of docked basal bodies in the RNAi animals (*Table 3*). Furthermore, we repeatedly noticed the striking occurrence of morphologically normal axonemes in the cytoplasm of the double knockdown animals (n = 29 in 94 μm of epithelial length examined for *Smed-phm+pam(RNAi)* vs n = 0 in 145 μm of epithelial length examined for control animals); these cytoplasmic axonemes were not surrounded by a membrane and were never observed in any of the other RNAi or L4440 control samples (*Figure 5H–K* and *Table 3*). Intriguingly, these mislocalized cytosolic axonemes appeared to have a normal 9 + 2 ultrastructure containing inner and outer dynein arms (*Figure 5K*) and were in general oriented along the long axis of the animal with the axonemal distal tip towards the head. The ectopic localization of axonemes in the cytoplasm arising due to PAM deficiency suggests a defect in basal body docking and/or ciliary membrane trafficking.

## PAM is required for primary cilia assembly in the developing neuroepithelium of mammals

Deletion of the *Pam* gene in mice results in embryonic lethality; embryos display massive pericardial edema and die by E14.5-E15.5 (*Czyzyk et al., 2005*). The cause and source of

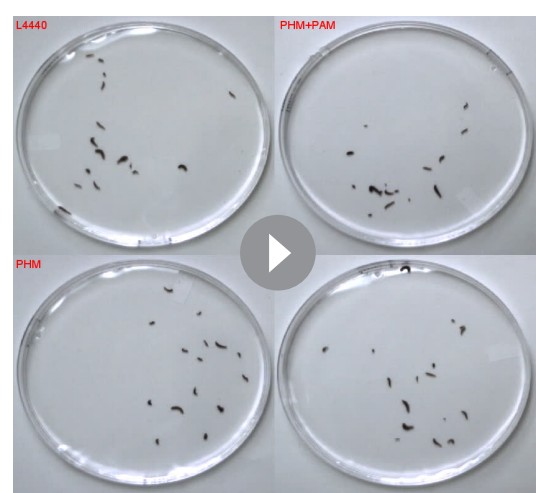

**Video 1.** Gliding Motility of Control and RNAi Planaria. Combined videos showing the movement of control (L4440), *Smed-phm+pam(RNAi)*, *Smed-phm(RNAi)* and *Smed-pam(RNAi)* animals in 9 cm Petri dishes. Videos were captured at 15 fps for 60 secs and play back at 10× real-time.

**Table 2.** Quantification of planarian motility. The table indicates the % immotile animals observed and gliding velocity and ciliary beat frequency for control and *Smed-pam(RNAi)*, *Smed-phm(RNAi)* and *Smed-phm+pam(RNAi)* planaria. Knockdown of PHM alone or PHM +PAM resulted in significant decreases in both gliding velocity and beat frequency.

| Planaria sample | % immotile animals | Gliding velocity in mm/sec (mean ± SEM) | Ciliary beat frequency (Hz) |
|---|---|---|---|
| L4440 | 10 | 0.70 ± 0.08 | 26 |
| PHM+PAM | 27 | 0.43 ± 0.06 | 17 |
| PHM | 32 | 0.37 ± 0.07 | 18 |
| PAM | 17 | 0.76 ± 0.09 | 30 |

this edema in $Pam^{-/-}$ embryos is unclear, but zebrafish and rodent ciliary mutants display a similar edematous phenotype (*Gorivodsky et al., 2009*; *Lee et al., 2015*; *Li et al., 2016b*; *Zhang et al., 2012*). To determine if cilia were defective in $Pam^{-/-}$ mice, we analyzed the neuroepithelium of E12.5 wild-type and knockout littermates by scanning electron microscopy, as these cells each possess a singular, primary cilium (*Louvi and Grove, 2011*; *Paridaen et al., 2015*). As expected, we observed primary cilia emerging from ciliary pockets in epithelial cells in the lateral ventricular surface of wild type embryos. Cilia were also present in $Pam^{-/-}$ neuroepithelial cells, although their morphology differed from cilia in wild-type animals (*Figure 6A*). Ciliary density was unaffected in the $Pam^{-/-}$ embryos (*Figure 6B*), but the cilia in these embryos were significantly shorter than controls (*Figure 6C*) (0.66 ± 0.01 μm in WT vs. 0.55 ± 0.01 μm in $Pam^{-/-}$ animals; n > 500). Since the longer control cilia were often bent, we also used transmission electron microscopy to confirm the presence of stumpy cilia in the $PAM^{-/-}$ animals; cilia measured 0.51 ± 0.04 μm (n = 69) in $Pam^{-/-}$ vs. 0.90 ± 0.08 μm (n = 45) in WT (*Figure 6D and E*). These combined data from three very different model organisms clearly reveal that PAM has a key and highly conserved role in ciliogenesis.

## PAM deficiency affects trafficking and transition zone proteins in *C. reinhardtii*

We next used *C. reinhardtii* to evaluate the effect of loss of PAM on levels of key ciliary and membrane trafficking proteins. As demonstrated in *Figure 1B*, lysates prepared from PAM amiRNA lines #3 and #8 had reduced amounts of CrPAM. Although total α-tubulin levels were unaltered in PAM amiRNA cells, levels of acetylated tubulin and poly-glutamylated tubulin, modified forms present in cilia, were both significantly lower compared to control cell lysates, consistent with the loss of cilia (*Figure 7A,B*).

To assess the impact of PAM deficiency on membrane trafficking, we assessed levels of Arf1, a Golgi-localized GTPase implicated in vesicular trafficking (*Hummel et al., 2007*), and the vesicular coat protein, clathrin. Clathrin and AP1 have been implicated in ciliogenesis and targeting of cargo proteins to the cilium (*Kaplan et al., 2010*) and the μ1A subunit of AP1 interacts with the cytosolic domain of rat PAM (*Bonnemaison et al., 2014*). Arf1 levels were unchanged in lysates of PAM knockdown cells, while clathrin levels were increased compared to control lysates (*Figure 7C,D*). This result suggested that certain membrane trafficking processes involving clathrin were affected in

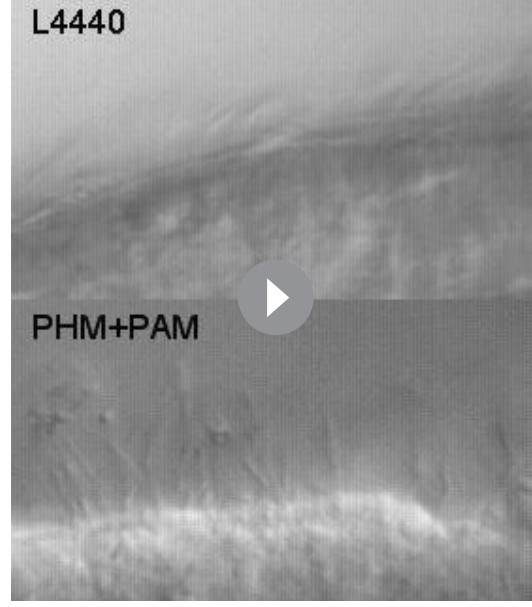

**Video 2.** Ciliary Motility of Control and *Smed-phm +pam(RNAi)* Planaria. Videos of ventral cilia of control (L4440) and *Smed-phm+pam(RNAi)* planaria were taken at 150 fps and play back at 1/10[th] real-time.

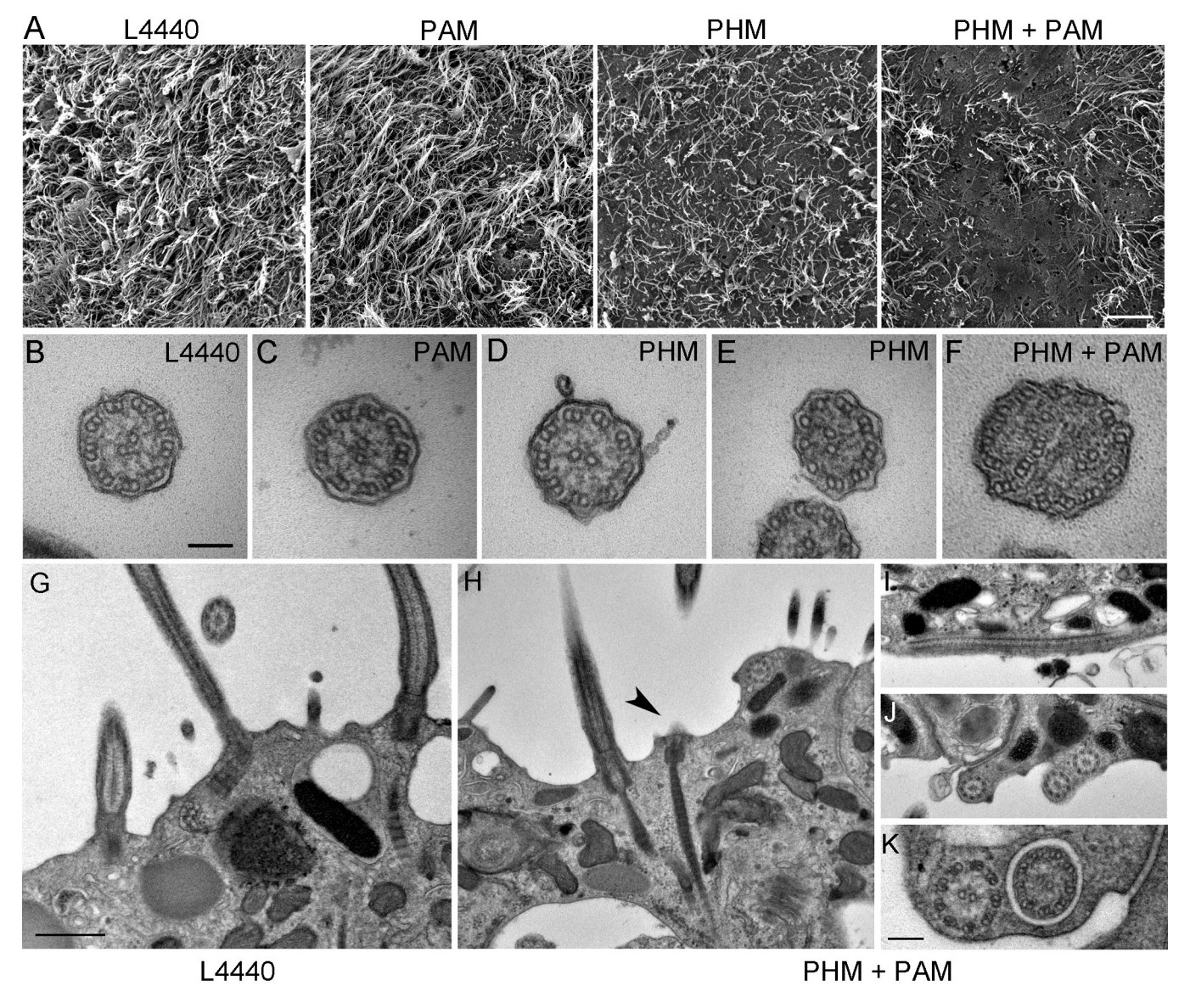

**Figure 5.** Ciliary morphology is affected in *Smed-phm(RNAi)* and *Smed-phm+pam(RNAi)* planaria. (A) Scanning electron micrographs of the ventral surface of an L4440-fed control planarian showed tightly packed cilia, while ciliary density was reduced in *Smed-phm(RNAi)* and *Smed-phm(RNAi)* +*Smed-pam(RNAi)* planaria; short ciliary stubs were visible on the ventral surface. *Smed-pam(RNAi)* animals displayed a modest, intermediate phenotype. Scale bar, 10 μm. (B–D) Transmission electron micrographs of ciliary cross sections showed normal 9 + 2 axonemal architecture in L4440, *Smed-pam(RNAi)* and *Smed-phm(RNAi)* planaria. (E–F) Singlet and doublet microtubules visible in the same plane in a *Smed-phm(RNAi)* cilium and abnormal, disorganized doublet microtubules in a *Smed-phm+pam(RNAi)* planarian cilium. Scale bar, 100 nm. (G) Transmission electron micrographs of ciliated ventral epithelium of L4440 and *Smed-phm+pam(RNAi)* (H–K) animals. Normal basal body docking and microtubule extension was visible in L4440; scale bar, 500 nm. In the double knockdowns, ciliary stubs (arrowhead, (H) and numerous cytosolic axonemes were visible (H–K). (I) Longitudinal section of a cytosolic axoneme parallel to the plasma membrane. (J) Cytosolic axonemes cut in cross section and all oriented with their distal tips towards the head of the animal. (K) A cytosolic axoneme next to a cilium contained in a ciliary pocket (right). Scale bar, 250 nm.

PAM amiRNA cells, while the contractile vacuole cycle, another membrane trafficking process, was unaltered (*Figure 1—figure supplement 2B and C*).

Our inability to observe Y linkers and the abnormal accumulation of microtubules and amorphous material in the short ciliary stubs seen in PAM amiRNA cells pointed to a dysfunctional transition zone; this zone regulates protein entry into the cilium and is essential for ciliogenesis (*Figure 7E*).

**Table 3.** Quantification of basal bodies and cytosolic axonemes in control and *Smed-phm+pam(RNAi)* planaria. The number of basal bodies docked at the cell surface with or without a ciliary extension and the number of observed cytosolic axonemes was determined for the indicated length of the ventral epithelium by TEM analysis. Numerous cytosolic axonemes were observed in PHM+PAM knock-down planaria, but none were found in controls.

| Planaria sample | Length scanned (μm) | # Docked Basal Bodies | | # Cytosolic Axonemes |
|---|---|---|---|---|
| | | + ciliary extension | No ciliary extension | |
| L4440 | 145 | 45 | 1 | 0 |
| PHM+PAM | 94 | 17 | 4 | 29 |

Levels of two transition zone components, CEP290 and NPHP4 (*Awata et al., 2014*) were monitored. Both PAM amiRNA strains contained greatly increased amounts of CEP290 and NPHP4 compared to controls (*Figure 7F,G*). CEP290 is a component of the Y linkers and *C. reinhardtii* CEP290 null mutants have short stumpy cilia (*Craige et al., 2010*) much like those observed in the PAM knockdown cells. NPHP4, a more distal component of the transition zone, is an essential part of the barrier at the ciliary base (*Awata et al., 2014*). It is possible that cells respond to an assembly defect caused by the reduced levels of CrPAM with a compensatory up-regulation of specific transition zone proteins. Interestingly, RNA-seq analysis revealed that the transcript abundance of these transition zone components was unchanged in PAM amiRNA cells (*Supplementary file 1*), pointing to a post-transcriptional regulatory response that alters CEP290 and NPHP4 protein levels in the absence of PAM.

To assess whether Golgi disruption caused the same effects on tubulin post-translational modification and transition zone protein expression as observed in the PAM-amiRNA strain, we treated control ciliated cells with brefeldin A (BFA) for 3 hr and observed the expected decrease in ciliary length (*Dentler, 2013*). The levels of acetylated and glutamylated tubulin were decreased in both PAM-amiRNA and brefeldin A-treated cells, consistent with loss of cilia. However, enhanced levels of the transition zone protein NPHP4 were not observed following brefeldin A treatment, indicating that this response is specific to PAM deficiency (*Figure 7—figure supplement 1*). Brefeldin A treatment increased Arf1 levels, a response not observed in the PAM-amiRNA strains. Taken together, these data indicate that the PAM knockdown phenotype is distinct from a generalized Golgi dysfunction phenotype.

To evaluate the localization of transition zone components in PAM amiRNA cells, we performed double labeling of cells with antibodies to CEP290 and acetylated tubulin. In control cells, punctate CEP290 staining was apparent at the base of each cilium (*Figure 7H* upper panels, enlargements on right). In PAM amiRNA cells, the localization pattern of CEP290 remained unchanged in the basal body region, which was identified by using antiserum to acetylated tubulin to locate the anterior portion of the cortical microtubules (*Figure 7H*, lower panels). However, additional CEP290 immunoreactive puncta were visible in the cell body (*Figure 7H*, arrows). These alterations in the levels of specific transition zone and membrane trafficking proteins, which are critical components of the ciliary assembly machinery, may contribute to the defects observed in PAM amiRNA *C. reinhardtii* cells.

## PAM depletion alters levels and localization of IFT proteins in *C. reinhardtii*

Both ultrastructurally and phenotypically, the responses observed following loss of PAM and loss of IFT proteins were quite similar. In *Chlamydomonas*, the anterograde motor kinesin-II and IFT-B components ferry cargo to the tip of the cilium (*Figure 7E*) (*Lechtreck, 2015*; *Taschner and Lorentzen, 2016*). Both RNA-seq analysis and immunoblotting showed that the levels of core anterograde IFT-B components, such as IFT46 and IFT72/74, were unaltered in PAM knockdown cells (*Supplementary file 1* and *Figure 8A,B*). Multiple bands observed with the IFT46 antibody may be due to a change in the phosphorylation state of the protein as has been observed previously (*Hou et al., 2007*); phosphatase treatment suggested that its phosphorylation status was altered in PAM amiRNA cells. IFT81 levels were increased in CrPAM amiRNA strain #8 but not in strain #3

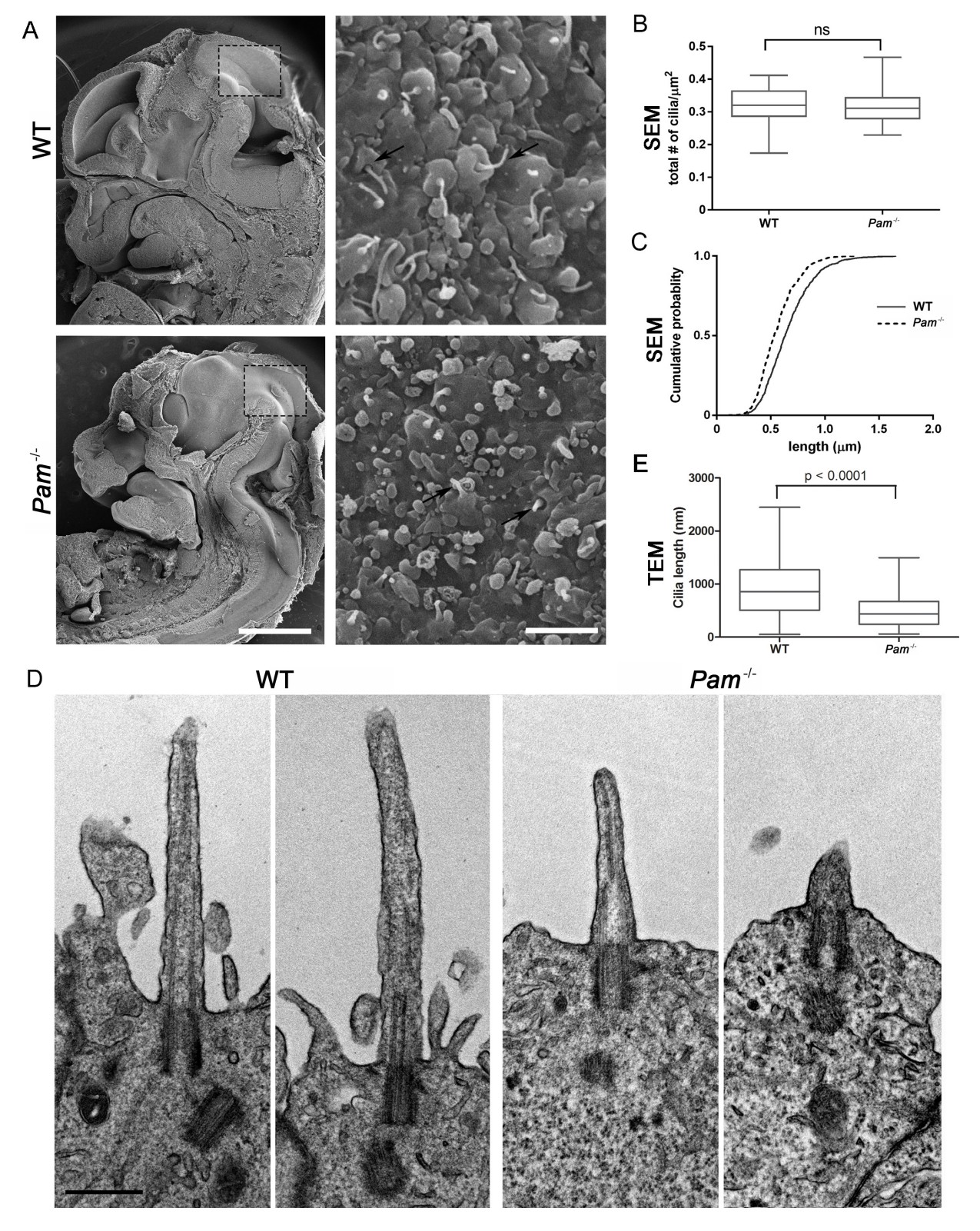

**Figure 6.** Primary cilia are shorter in *Pam*⁻/⁻ neuroepithelial cells. (**A**) Low magnification scanning electron micrographs of longitudinal sections through E12.5 WT and *Pam*⁻/⁻ mouse embryos are on the left. Scale bar, 1 mm. The panels on the right show neuroepithelial cell cilia on the ventricular surface of WT and *Pam*⁻/⁻ embryos. Scale bar, 2 μm. (**B**) Ciliary density (number of cilia/μm², measured from SEM images) was not significantly different between WT and PAM⁻/⁻. (**C**) Cumulative distribution of ciliary lengths measured from SEM images differed in *Pam*⁻/⁻ vs WT control (Kolmogorov-Smirnov test, *Figure 6 continued on next page*

Figure 6 continued

p<0.0001 for n > 500 measurements). Mean ciliary length in WT was 0.66 ± 0.01 μm (mean ± SEM) and in *Pam⁻/⁻* was 0.55 ± 0.01 μm. (**D**) Representative transmission electron micrographs from E12.5 WT and PAM⁻/⁻ mouse embryo neuroepithelium. Scale bar, 500 nm. (**E**) Cilia (measured from TEM images) were significantly shorter (p<0.0001, unpaired t-test) in the *Pam⁻/⁻* neuroepithelial cells (0.51 ± 0.04 μm, n = 69) compared to WT cells (0.90 ± 0.08 μm, n = 45).

(*Figure 8A,B*). Since, strain #8 had lower amounts of CrPAM than strain #3 (*Figure 7A,B*), IFT81 levels may be sensitive only to a severe deficiency of CrPAM.

IFT proteins normally display a punctate localization in the cilium and accumulate at its base, close to the basal body. Using antibodies to α-tubulin to mark the cilium, control cells exhibited the expected localization of IFT72/74. In PAM amiRNA cells, IFT72/74 staining was observed in the ciliary stubs (*Figure 8C*, insets) and staining in the peri-basal body region was reduced. To determine if these changes in localization were specific to IFT72/74, we double-labeled control and PAM amiRNA #8 cells with antibodies to IFT46 and IFT81. Staining for IFT46 and IFT81 showed significant overlap in the basal body region and cilium in control cells. In the PAM amiRNA cells, both IFT components were mostly located in the short ciliary stubs and the amount in the basal body region was reduced (*Figure 8D*). The abnormal accumulation of IFT components in the ciliary stubs may contribute to the increase in electron-dense amorphous material observed by electron microscopy (*Figure 3B–D*).

Strikingly, lysates of PAM amiRNA strains contained increased amounts of IFT-A component IFT139 compared to control cell lysates (*Figure 8E,F*). The retrograde IFT dynein motor works with IFT-A components to return proteins from the cilium to the cell body (*Figure 7E*) (*Lechtreck, 2015*; *Taschner and Lorentzen, 2016*). In *C. elegans,* loss of IFT 139 leads to formation of abnormal, short cilia (*Niwa, 2016*), and mutations in the human gene encoding IFT139 (TTC21B) have been linked to nephronophthisis and thoracic dystrophy (*Davis et al., 2011*; *Huynh Cong et al., 2014*). Levels of axonemal outer arm dynein intermediate chain (IC2) were unchanged in PAM amiRNA cell lysates (*Figure 8E,F*). Thus, changes in PAM levels affect specific components of the IFT machinery. This compensatory increase occurs post-transcriptionally (*Supplementary file 1*).

## PAM depletion affects selected aspects of post-Golgi trafficking in *C. reinhardtii* cells

Like rat PAM, CrPAM is largely localized to the Golgi region (*Kumar et al., 2016b*); the membrane proteins needed by the growing cilium are provided by the Golgi. We next asked if PAM deficiency affected the ultrastructure of this organelle. Transmission electron microscopy revealed subtle alterations in Golgi structure in PAM amiRNA strains #3 and #8 compared to two empty vector control strains (*Figure 9A*). The area occupied by the Golgi, number of cisternae and Golgi width were not significantly different in PAM amiRNA cells (*Figure 9B,C,D*). However, a significant difference in Golgi curvature was observed (*Figure 9E*); Golgi stacks in the knockdown cells were more concave.

Starch grains, which are present in the chloroplast, accumulate when Golgi function is inhibited (*Hummel et al., 2010*). Since we observed ultrastructural changes in the Golgi complex and changes in clathrin levels, we examined plastid starch grains in PAM amiRNA and control cells by electron microscopy (arrows in *Figure 10A*). Quantification revealed that starch grains occupied a larger fraction of the cell area in PAM knockdown cells compared to control cells; cell size did not differ (*Figure 10B,C*). To determine whether this morphological change in starch grains correlated with an increase in starch content in the PAM amiRNA cells, we measured starch levels enzymatically and found no significant difference between the knockdown and control cells. Enlarged starch grains without a concomitant increase in starch levels were previously noted in vacuolar membrane protein-1 (VMP1) knockdown strains, which also displayed abnormal Golgi morphology (*Tenenboim et al., 2014*). These results suggest an impact of PAM knockdown on Golgi function.

To further assess alterations in Golgi function, empty vector and PAM amiRNA cells were cultured under nutrient deficiency conditions known to alter secretion of specific proteins. Iron deficiency conditions (0.5 μM vs 20 μM Fe) stimulate the secretion of FEA1, an iron assimilation protein that is readily identifiable after electrophoretic separation of the spent medium, which in cell wall-less strains contains the secreted protein (*Allen et al., 2007*). Spent medium normalized to equal cell protein was analyzed by silver staining. As expected, the 40 kDa FEA1 protein was readily detected

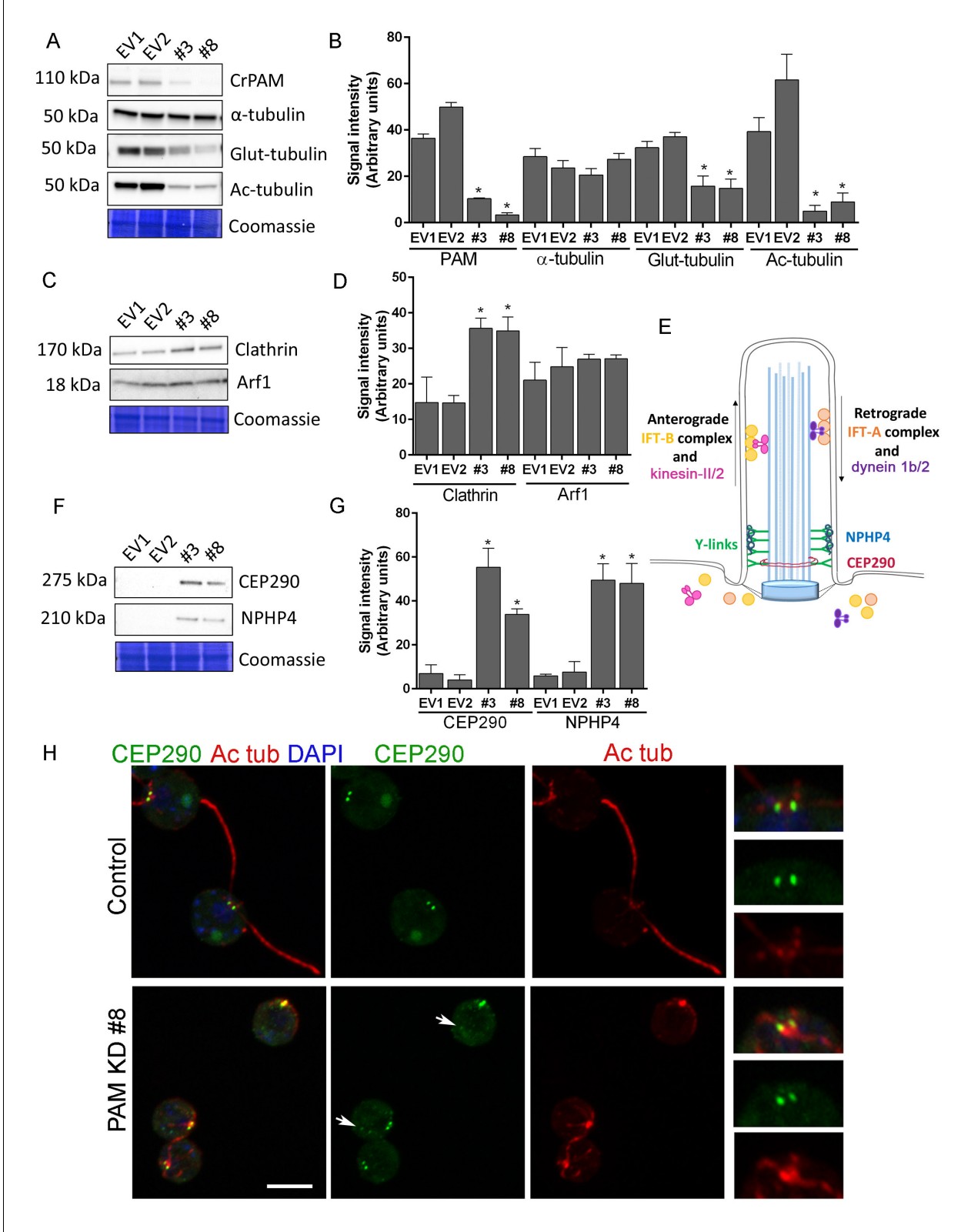

**Figure 7.** Knockdown of CrPAM alters levels and localization of transition zone proteins. At least three lysates of EV1, EV2, #3 and #8 cells were subjected to western blot analysis. Representative blots and Coomassie-stained gels are shown (**A, C, F**); bar graphs indicate mean normalized intensity from all analyses +/- SEM (**B, D, G**). Statistical analysis was performed using one-way Anova. * indicates p<0.05 for both EV strains vs both PAM amiRNA strains. (**A, B**) Antibodies to CrPAM and modified forms of tubulin were used; acetylated and poly-glutamylated tubulin levels were lower in

*Figure 7 continued on next page*

*Figure 7 continued*

PAM amiRNA strains, but levels of α-tubulin were unchanged. (**C**, **D**) Antisera to clathrin heavy chain and Arf1 were used; PAM amiRNA strains had higher levels of clathrin heavy chain; Arf1 levels were not different. (**E**) Schematic of IFT and transition zone components in cilia; for the IFT motors, both *Chlamydomonas* and mammalian nomenclatures are shown. (**F**, **G**) Antisera to transition zone components CEP290 and NPHP4 were used; levels of both transition zone components were increased in PAM amiRNA mutant strains. (**H**) Maximum projection of optical sections of EV and PAM amiRNA *C. reinhardtii* cells labeled with antibodies to CEP290 (green) and acetylated tubulin (Ac tub, red). Merged images also show DAPI (blue), to locate the nucleus. Arrows point to CEP290 foci in the cell body. Enlargements at right show CEP290 localization at the base of cilia.

The following figure supplement is available for figure 7:

**Figure supplement 1.** Effect of Golgi disruption by brefeldin A on tubulin post-translational modification and transition zone protein levels.

in all strains under iron deficiency conditions (*Figure 10D*). However, secretion of FEA1 was decreased in the PAM amiRNA strains compared to controls. Furthermore, there were striking differences in the secretome in PAM knockdown cells even under basal conditions (arrowheads and brackets, *Figure 10C*). Surprisingly, FEA1 transcript levels were two-fold higher in PAM amiRNA cells than in controls, perhaps reflecting a compensatory measure (*Supplementary file 1*). We next cultured cells under phosphate-deficient conditions, which stimulates secretion of alkaline phosphatase (*Quisel et al., 1996*) or sulfate-deficient conditions, which stimulates secretion of arylsulfatase (*de Hostos et al., 1988*). Cellular alkaline phosphatase activity levels were indistinguishable in phosphate-deficient control and PAM amiRNA cells; however, alkaline phosphatase secretion was greatly attenuated in PAM amiRNA strains compared to controls (*Figure 10E*). In contrast, secretion of arylsulfatase by sulfate-deprived cells did not differ in PAM amiRNA and control cells (*Figure 10F*). Taken together, these data indicate that the effect of PAM loss on secretion is selective and not a result of generalized Golgi dysfunction.

## Discussion

### PAM plays a critical role in the assembly of motile and primary cilia

Our work has identified PAM as a crucial, evolutionarily conserved factor that is required for ciliogenesis in multiple cell types. PAM is important for the biogenesis of the motile cilia present in *C. reinhardtii* and on the ventral epithelium of planaria, as well as the normal assembly of the primary cilia of neuroepithelial cells in mammals. In *C. reinhardtii*, reduction of PAM did not alter cell growth rate but led to a dramatic loss of cilia, with assembly failing beyond the transition zone; the ciliary stubs contained amorphous electron dense material thought to consist of IFT particles and short pieces of microtubules at abnormal orientations (*Figure 3B–D*). Although basal body docking and cell division were comparable to control cells, extension of the microtubules from the basal body was disrupted in PAM amiRNA cells, similar to the phenotype of mutants defective for IFT88 and IFT46 (*Hou et al., 2007*; *Pazour et al., 2000*).

Analyzing the roles of monofunctional PHM and bifunctional membrane PAM in planaria revealed additional complexities in the regulation of ciliogenesis (*Figure 4*). Planarians progressively showed motility defects as the PAM genes were silenced over the three-week treatment. PHM activity was reduced to a similar extent in *Smed-phm(RNAi)* and *Smed-pam(RNAi)* animals and to a greater extent when expression of both genes was reduced. Our RNAi analysis revealed a predominant role for PHM in gliding velocity; simultaneous knockdown of soluble PHM and membrane PAM resulted in a more robust defect in ciliogenesis than knockdown of either single gene (*Figure 5A*). In the double knockdown animals, we observed a striking loss of motile cilia and the ectopic localization of fully assembled axonemes in the cytoplasm of cells in the ventral epithelium (*Figure 5H–K*). Thus, along with extension of the microtubules from basal bodies docked at the plasma membrane, PAM activity may be important for proper docking of the ciliary vesicle and orienting the basal body. To our knowledge, this is the first report of 'exposed' axonemes in the cytoplasm of multiciliated cells in metazoans. A few studies reported the occurrence of microtubules in the cytoplasm in the absence of ciliary genes, but it is not clear that these microtubules formed an axonemal structure (*Burke et al., 2014*; *Boisvieux-Ulrich et al., 1990*; *Park et al., 2006*). The complete lack of a ciliary

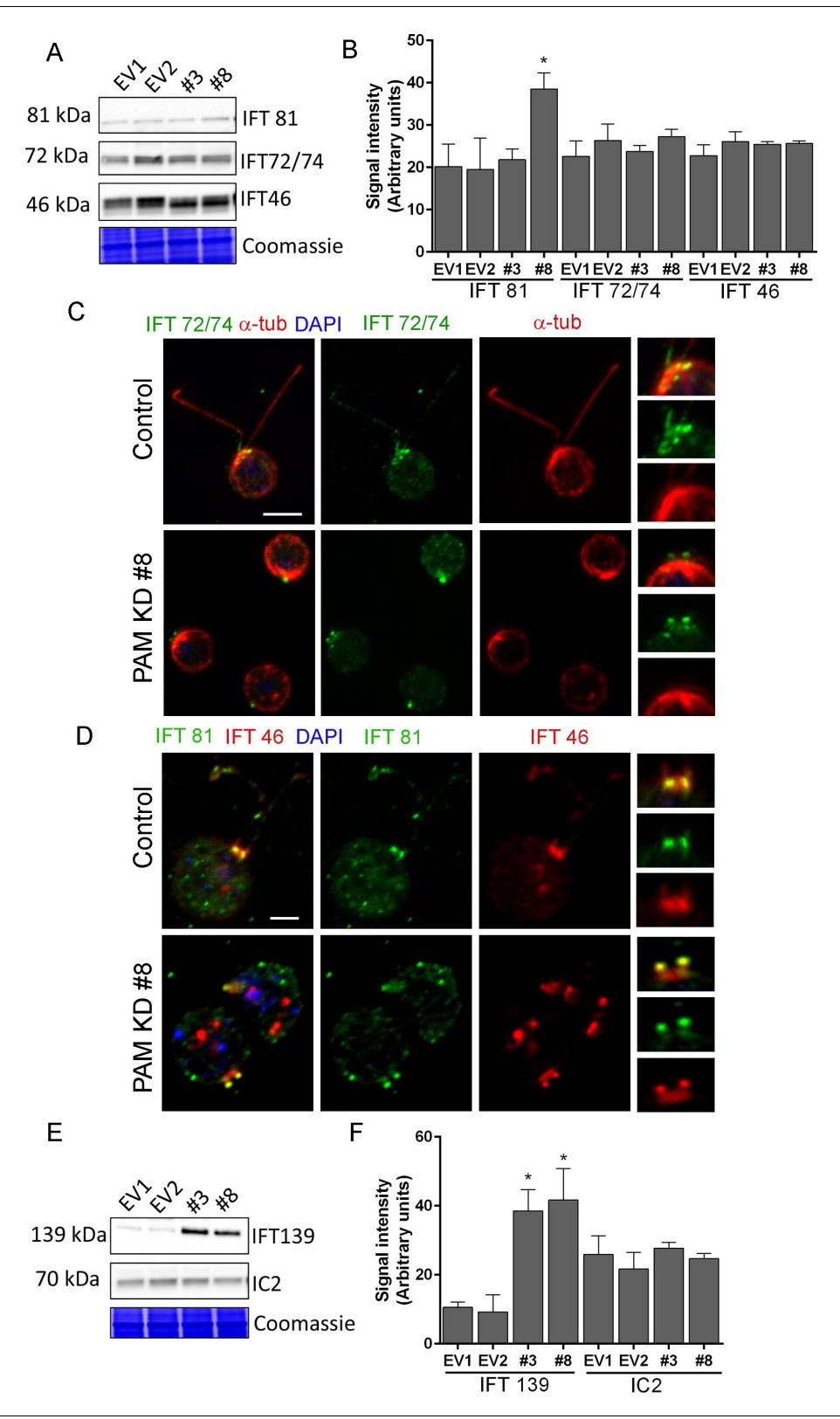

**Figure 8.** Knockdown of CrPAM alters levels and localization of selected IFT components. (A and B) *C. reinhardtii* EV and PAM amiRNA cell lysates immunoblotted with antibodies to IFT-B components, IFT81, IFT72/74 and IFT46 and quantified as described for *Figure 7*; levels of IFT81 were increased in PAM amiRNA #8 strain compared to EV controls. Representative Coomassie-stained gels show equal loading. (C) In control cells, IFT72/74 (green) was located at the base of the cilium (enlargement at right) and along its length (marked by antibodies to alpha-tubulin (red)). Images acquired

*Figure 8 continued on next page*

*Figure 8 continued*

at the same exposure settings showed IFT72/74 in the ciliary stubs and decreased immunofluorescence in the peri-basal body region (enlargement at right) in PAM amiRNA cells. Scale bar, 5 µm. (D) *C. reinhardtii* control and PAM amiRNA #8 cells co-stained with antibodies to IFT81 (green) and IFT46 (red). Images acquired at similar exposure settings showed redistribution of both proteins into ciliary stubs in PAM amiRNA cells (enlargements at right) compared to predominantly peri-basal body localization in controls. (E and F) Representative western blot of a retrograde ciliary transport protein; levels of IFT-A component IFT139, were higher in cell lysates from PAM amiRNA *C. reinhardtii* cells compared to EV control strains; axonemal outer arm dynein IC2 levels did not differ.

membrane around the apparently fully-assembled 9 + 2 axonemal structure raises the possibility that the supply of membrane to the growing cilium is a key limiting factor in the *Smed-phm+pam (RNAi)* animals.

While reduction of PAM in *C. reinhardtii* and planaria resulted in a dramatic defect in ciliary assembly, complete loss of PAM in mice did not abolish primary cilia formation in neuroepithelial cells, although they were significantly shorter than cilia in wild type animals and in many cases only ciliary stubs were evident (*Figure 6*). It is unknown if these short cilia are functional.

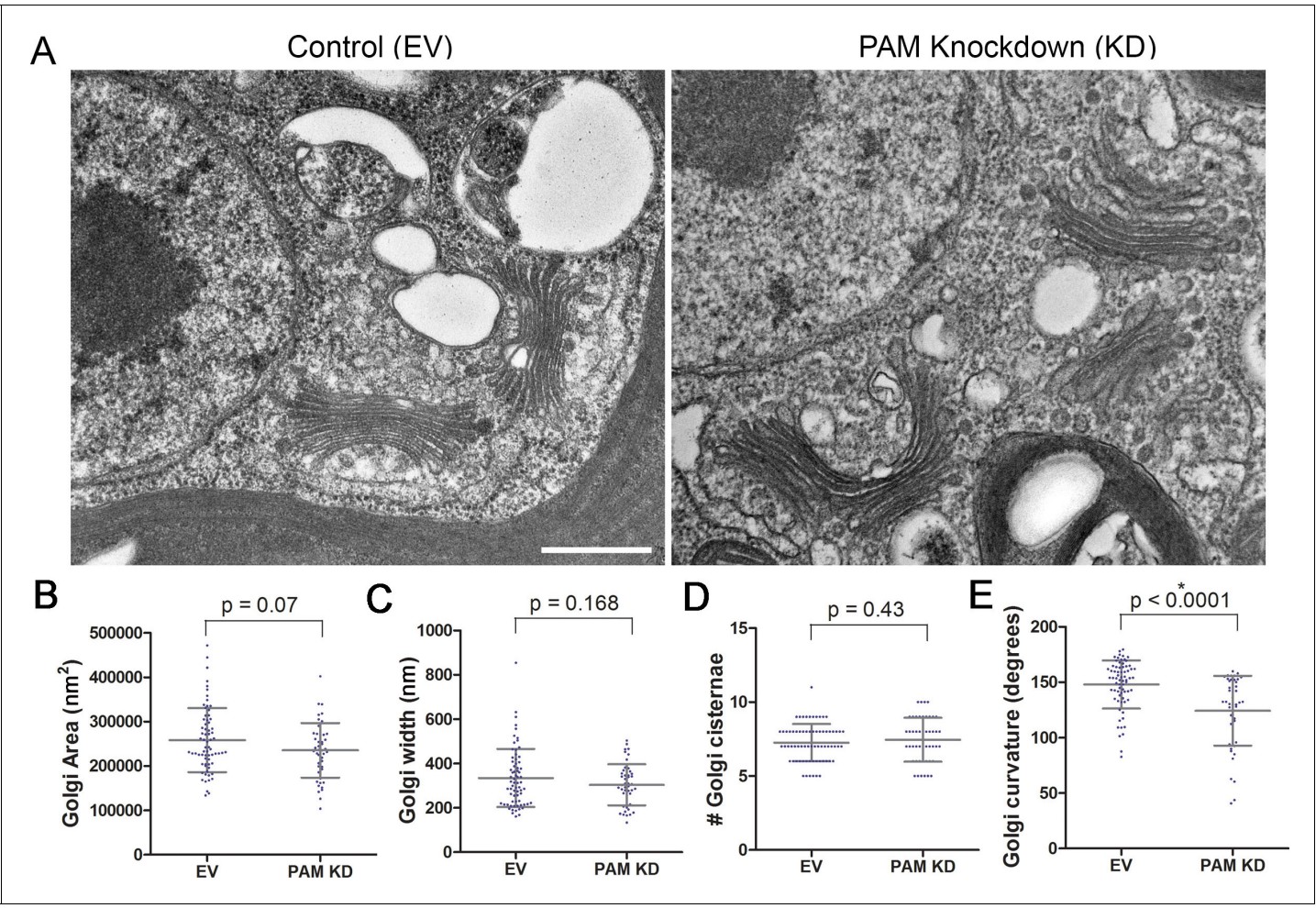

**Figure 9.** Golgi ultrastructure is altered in PAM amiRNA cells. (A) Transmission electron micrographs showing Golgi morphology in control and PAM amiRNA *C. reinhardtii* cells. Scale bar, 500 nm. Total area occupied by Golgi stacks (B), width of Golgi cisternae at the center of each stack (C) and number of Golgi cisternae (D) did not differ in PAM knockdown vs control cells. The degree of curvature of the Golgi cisternae from the center of the stack to the tip was significantly lower in PAM knockdown cells compared to controls (E). For all graphs, data were plotted as a scatter dot plot with mean ± SD values; unpaired t-test was used for statistical analysis in each case.

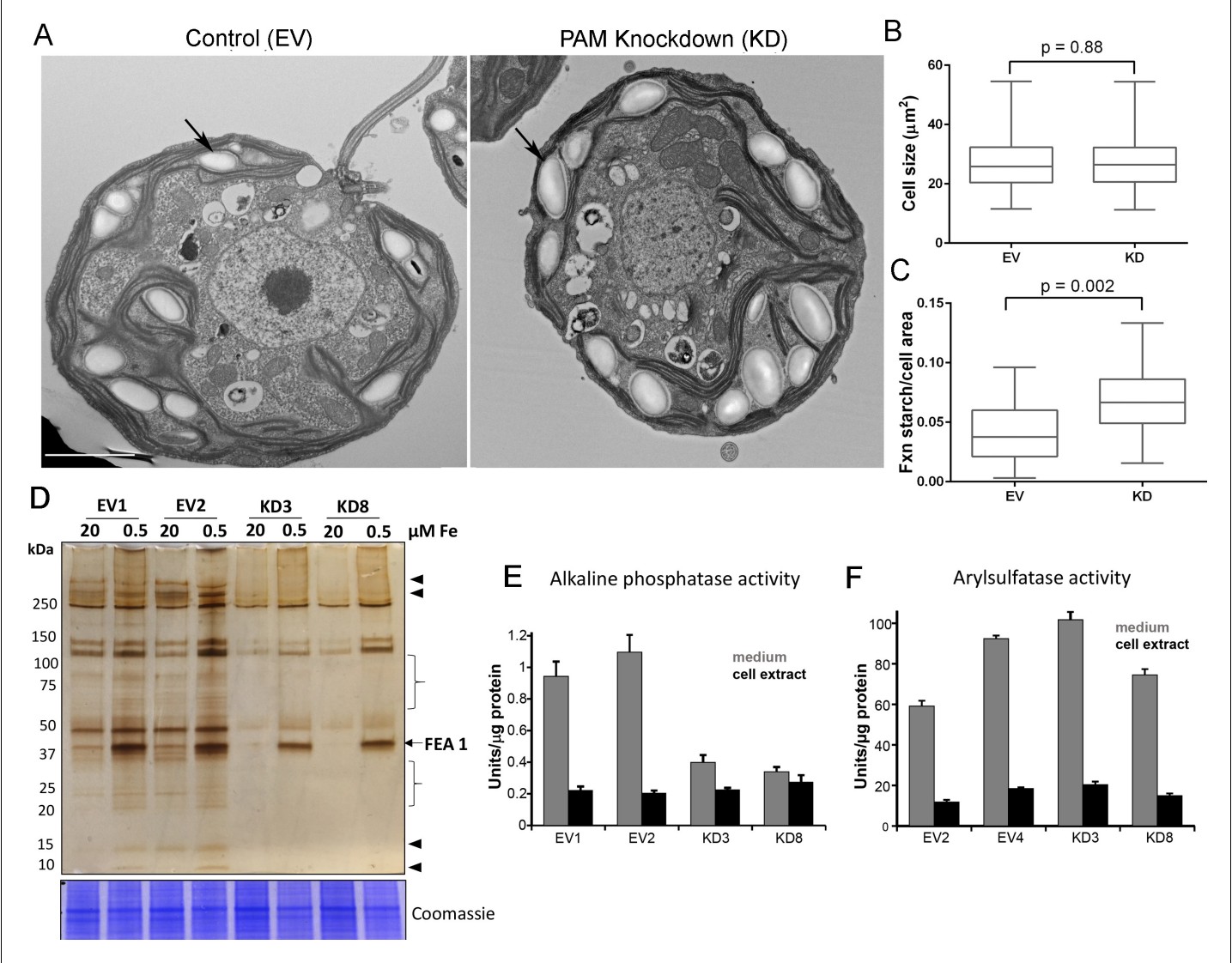

**Figure 10.** Golgi function is selectively compromised in *C. reinhardtii* PAM knockdown cells. (**A**) Transmission electron micrographs showing enlarged starch grains in a representative PAM amiRNA cell compared to a control cell. Scale bar, 2 μm. (**B**) Control and PAM amiRNA cells did not differ in cell size (n > 500 measurements from scanning electron micrographs) (**C**) Starch grains occupied a larger fraction of cell area in PAM amiRNA cells compared to controls (n ~20 measurements from TEM images). Unpaired t-test was used to analyze values in box and whisker plots. (**D**) Control and PAM amiRNA cells cultured in normal (20 μM Fe) and iron deficiency (0.5 μM Fe) conditions showed altered secretion of FEA1 and many other proteins (arrowheads and braces). For each strain, spent medium from an equal amount of total cell protein (verified by Coomassie staining of cell lysates) was analyzed by silver staining. (**E**) Alkaline phosphatase activity (units/μg cell protein) was assessed in cell lysates and spent media (collected after 24 hr) from two different phosphate-deprived PAM amiRNA and control strains. one unit = 1 nmole/h. (**F**) Arylsulfatase activity (units/μg cell protein) was assessed in cell lysates and spent media (collected after 24 hr) from two different sulfate-deprived PAM amiRNA and control strains. one unit = 1 nmole/h.

The following figure supplement is available for figure 10:

**Figure supplement 1.** Model illustrating how loss of PAM affects ciliogenesis.

Why does loss of PAM lead to closely related, yet pleomorphic defects in ciliogenesis in different organisms? The answer is unclear, but differences in the pathways utilized for assembling cilia in various cell types may contribute to the different phenotypes observed. Indeed, distinct cell- and organism-specific phenotypes have been observed when ciliary genes are mutated (*Chaya et al., 2014*;

*Lechtreck et al., 2008*; *Lechtreck and Witman, 2007*). Multiciliated epithelial cells and *C. reinhardtii* cells utilize an extracellular pathway of ciliogenesis, where the basal body first docks at a distinct region of the plasma membrane and ciliary extension into the extracellular environment is a result of axonemal growth. In contrast, fibroblasts and neuroepithelial cells employ an intracellular pathway of ciliogenesis (*Molla-Herman et al., 2010*), where growth of the axoneme is initiated inside a ciliary vesicle in the cytoplasm. Docking of the ciliary vesicle subsequently leads to emergence of the cilium outward from the cell surface. Key steps in these ciliogenic pathways may exhibit differential sensitivity to the lack of PAM in distinct organisms and tissues.

## PAM deficiency alters Golgi ultrastructure, function and post-Golgi trafficking

The Golgi serves as a sorting station for proteins targeted to the secretory pathway, endosomes and lysosomes and the Golgi region is the main site of PAM localization in *C. reinhardtii* and mammalian cells (*Kumar et al., 2016b*); after exocytosis, mammalian PAM traverses the endocytic pathway, returning to the trans-Golgi network. The Golgi supplies membrane proteins required for ciliary expansion; vesicles carrying ciliary cargo must be assembled and trafficked to the growing ciliary base. Loss of PAM in *C. reinhardtii* altered Golgi architecture; Golgi stacks were more curved in PAM amiRNA cells than in control cells (*Figure 10A,E*). Golgi morphology is affected by the cytoskeleton and Golgi matrix, and mechanical and structural alterations to the Golgi are known to affect the formation of vesicles and tubules (*Guet et al., 2014*). In agreement with alterations in Golgi function, PAM amiRNA *C. reinhardtii* cells had enlarged starch grains, a response known to reflect defects in the secretory pathway and the trafficking of metabolic enzymes (*Hummel et al., 2010*). Although, we observed defects in secretion of FEA1 (*Figure 10D*) and alkaline phosphatase (*Figure 10E*), arylsulfatase secretion (*Figure 10F*) and contractile vacuole function (*Figure 1—figure supplement 2B,C*) were unaffected in PAM knockdown strains, pointing to selective, rather than global, changes in vesicular trafficking.

The primarily Golgi-localized AP1 complex, known to be important for bi-directional trafficking of proteins between endosomes, secretory granules and the trans-Golgi network, has also been implicated in the ciliary trafficking of proteins (*Kaplan et al., 2010*; *Dwyer et al., 2001*). The μ1A subunit of AP1 interacts with the cytosolic domain of PAM; reducing μ1A expression impairs the formation of mature, secretagogue-responsive secretory granules (*Bonnemaison et al., 2014*). Mature secretory vesicles lack clathrin; clathrin-coated vesicles bud from immature secretory granules, which are localized near the trans-Golgi network. Clathrin levels were almost double that of control cells in *C. reinhardtii* PAM amiRNA strains, again suggesting that aspects of vesicular trafficking are abnormal in these cells. Thus, alterations in post-Golgi trafficking to the cilium might contribute to defects in ciliogenesis in PAM-deficient cells.

Although the levels of most of the IFT proteins analyzed were unchanged, PAM amiRNA cells displayed a striking increase in the retrograde trafficking IFT-A component, IFT139. IFT components normally assemble into IFT trains at the base of the cilium, displaying a peri-basal body distribution along with punctate staining in the cilium. In PAM amiRNA cells, several IFT-B components accumulated in the ciliary stubs and staining at the base of the stubs was decreased, suggesting disrupted transition zone function. The transition zone at the proximal end of the cilium is a site of protein sorting and trafficking that is critical for ciliogenesis. Mutations in transition zone components lead to ciliopathies such as Meckel-Gruber and Joubert syndromes. We were unable to identify distinct Y linkers in our PAM amiRNA *C. reinhardtii* cells. Strikingly, although levels of transcripts encoding CEP290 and NPHP4 were unaltered in PAM amiRNA knockdown cells, levels of CEP290 protein, a component of the Y linkers and NPHP4 protein, a distal transition zone component, were elevated substantially. Furthermore, Golgi disruption by brefeldin A did not cause a similar increase in NPHP4 levels, pointing to a PAM-specific effect distinct from general Golgi dysfunction (*Figure 7—figure supplement 1*). CEP290 still localized to the base of the cilium in PAM amiRNA cells, but additional foci of CEP290 staining were visible elsewhere in the cell. Together, our biochemical and ultrastructural data suggest that transition zone function is compromised in PAM amiRNA mutant cells. Post-transcriptional control mechanisms appear to play an important role in these responses, since these changes in IFT and transition zone protein levels do not occur at the transcriptional level (*Supplementary file 1*), suggesting that there may not be a simple direct connection between the expression of known ciliary proteins and PAM. Post-transcriptional or protein stability changes that

contribute to variations in the levels of these proteins in the absence of PAM need further investigation.

## Possible mechanisms for the effects of PAM on ciliary assembly

Given the recognized roles of PAM in membrane trafficking mediated by its cytosolic domain, a key question is whether it is PAM protein per se or its enzymatic (amidating) activity that is the key factor in ciliogenesis (*Figure 10—figure supplement 1*). Currently, four results point to a requirement for active PAM enzyme and an amidated product in ciliary assembly. First, *C. reinhardtii* CrPAM-ΔCD cells, with a PAM protein that lacks most of its C-terminal domain, are capable of assembling a cilium. Second, treatment of wildtype *C. reinhardtii* cells with a mechanism-based PHM inhibitor (4-phenyl-3-butenoic acid, PBA) delayed ciliary regrowth following deciliation. Third, mammalian PHM has a relatively low affinity for copper, and copper chelation is routinely used to inhibit PHM (*Mains et al., 1986*). Treatment of *C. reinhardtii* cells with the copper-selective chelator neocuproine mimics the effects of PBA on reciliation. Finally, knockdown of the *Smed-phm* gene, which encodes soluble PHM, alters planarian cilium biogenesis; the bifunctional *Smed-pam* gene is unable to compensate for its loss.

How might amidation influence cilium assembly? While a secreted, bioactive peptide product may act as a positive regulator of ciliogenesis, glycine-extended lipids are also PAM substrates (*Merkler et al., 2004*). The synthesis of oleamide, an amidated lipid, in a mouse neuroblastoma cell line was recently shown to require PAM; siRNA mediated knockdown of PAM resulted in accumulation of the N-acylglycine precursor (*Jeffries et al., 2016*). Amidated lipids produced by PAM could generate lipid sub-domains in the secretory pathway and serve to enrich secretory pathway localized proteins into specialized vesicles. In support of this model, we find PAM loss selectively alters post-Golgi trafficking in *C. reinhardtii*. Furthermore, doxycycline-induced expression of membrane PAM in mouse pituitary tumor cells has a profound effect on post-Golgi trafficking (*Ciccotosto et al., 1999*). Soluble cargo proteins are diverted away from the regulated secretory pathway and constitutive secretion is increased. This effect is attributed, in part, to striking changes in the organization of the actin cytoskeleton (*Ciccotosto et al., 1999*), which may be mediated through interactions of the cytosolic domain of PAM with actin regulatory proteins.

A connection between actin and PAM is particularly intriguing since actin is required for basal body orientation and docking in multi-ciliated cells (*Boisvieux-Ulrich et al., 1990*; *Park et al., 2006*; *Pan et al., 2007*; *Meunier and Azimzadeh, 2016*; *Park et al., 2008*). In *C. reinhardtii*, conventional actin is found near the Golgi complex and in ciliary inner arm dynein complexes (*Kato-Minoura et al., 1997*; *Piperno and Luck, 1981*), and participates in linking ciliary assembly to ciliary length (*Avasthi et al., 2014*). Pharmacological studies and unbiased screens have revealed complex roles for the actin cytoskeleton in controlling primary ciliogenesis and function (*Kim et al., 2010*, *2015*; *Phua et al., 2017*; *Nager et al., 2017*; *Cao et al., 2012*). The small pool of PAM located in the cilium may also be essential for ciliogenesis. Identifying the domains/regions of PAM necessary for its localization to the cilium will be important to address this issue.

Our PBA and neocuproine studies indicate that wildtype cells can regrow cilia even when PHM activity has been inhibited. Thus, it is possible that cilia can regrow in the absence of enzymatically active PAM provided that all the necessary structures (such as attached basal bodies, functional IFT machinery *etc*) are already available. However, as *Chlamydomonas* stockpiles sufficient components to regenerate two approximately half-length cilia in the absence of additional protein synthesis (*Rosenbaum et al., 1969*), it is also possible that these cells, which initially contain wild type levels of enzymatically active PAM, already have enough amidated products available to allow ciliogenesis to occur, albeit in a delayed manner.

The evolutionarily widespread occurrence of the PAM gene and the results presented here suggest that at least one of the ancestral and conserved roles of PAM is in ciliary assembly. The prevailing theory of ciliary evolution is that polarized trafficking of signaling proteins and receptors from the Golgi complex using coat proteins, such as IFT and COP proteins, gave rise to a plasma membrane sensory patch that ultimately gained a microtubule backbone (*Carvalho-Santos et al., 2011*; *Jekley Jékely and Arendt, 2006*). It is possible that PAM was included in the set of signaling proteins trafficked to the ciliary compartment from the Golgi in the ancestral eukaryote. In conclusion, our results suggest that both the amidation and membrane trafficking roles of PAM contribute to ciliogenesis.

# Materials and methods

## *C. reinhardtii* strains and culture conditions

Two miRNAs targeting *C. reinhardtii CrPAM* were designed according to (*Molnar et al., 2009*) using the WMD3 tool at http://wmd3.weigelworld.org/. Resulting oligonucleotides for amiRNA1

### PAMamiFor1
ctagtAAGACGAGCCACCCCAATGTAtctcgctgatcggcaccatgggggtggtggtgatcagcgctaTACAATGGGGTGGCTCGTCTTg;

### PAMamiRev1
ctagcAAGACGAGCCACCCCATTGTAtagcgctgatcaccaccacccccatggtgccgatcagcgagaTACATTGGGGTGGCTCGTCTTa and amiRNA2

### PAMamiFor2
ctagtCCGGTTCTTGTTAGCTGCTCAtctcgctgatcggcaccatgggggtggtggtgatcagcgctaTGAGTGGCTAACAAGAACCGGg;

### PAMamiRev2
ctagcCCGGTTCTTGTTAGCCACTCAtagcgctgatcaccaccacccccatggtgccgatcagcgagaTGAGCAGCTAACAAGAACCGGa (uppercase letters representing miRNA*/miRNA sequences) were annealed by boiling and slowly cooling-down in a thermocycler and ligated into SpeI-digested pChlamiRNA2 (17). Plasmids were linearized by digestion with HindIII and transformed into *C. reinhardtii* strain CC-4351 cw15 mt+ by vortexing with glass beads (*Kindle, 1990*). The amiRNA2 knockdown strains were used for most experiments. The CC-4533 cw15 mt- control and CrPAM-*ΔCD* (LMJ.RY0402.20428) strains were procured from the indexed *C. reinhardtii* mutant library (*Li et al., 2016a*). Disruption of the C-terminal domain of CrPAM in CrPAM-*ΔCD* strain was verified by PCR using protocols described in (*Li et al., 2016a*). All strains where grown under constant illumination in TAP medium with revised trace elements (Special K) instead of Hutner´s trace elements according to (*Kropat et al., 2011*). The TAP medium was supplemented with 50 mg L$^{-1}$ of arginine when required. Growth curves were obtained from cells grown in both TAP and minimal media.

## Effect of PHM inhibitors on reciliation

To assess the effects of PHM inhibitors on ciliogenesis, CC-124 wildtype cells were pretreated for 3 hr with 200 µM 4-phenyl-3-butenoic acid (PBA; Sigma-Aldrich) or 10 µM neocuproine (Sigma-Aldrich) prior to deciliation by pH shock (*Craige et al., 2013*); cells treated with neocuproine were grown in a low copper medium, lacking added copper in the trace elements. Cells were fixed at various time points with 2% formaldehyde and ciliary length was measured.

## Planarian culture conditions, RNAi and motility assays

*S. mediterranea* were maintained in 1× solution of Montjuïc salts as described previously (*Rompolas et al., 2009*). Knockdown of PAM mediated by RNAi was accomplished by cloning 300 bp regions of *Smed-phm* (SMU15001346 in the SmedGD Genome database [*Robb et al., 2008*]) and *Smed-pam* (SMU15006556) into the L4440 plasmid as described previously (*Rompolas et al., 2009*, *2013*). Briefly, the constructs were transformed into HT115 (DE3) *E.coli* cells lacking RNase III to avoid degradation of dsRNA (*Newmark et al., 2003*). After induction of dsRNA with IPTG, bacterial pellets mixed with calf liver were fed to groups of planarians (~20 animals each set) twice a week. For knockdown of *Smed-phm+pam*, the bacterial pellets were mixed in a 1:1 ratio before feeding. Reduction of target gene mRNA expression was monitored 2–3 weeks post feeding by RT-PCR using the same primers used to generate the RNAi vectors.

Planarian gliding motility was monitored during week three by recording their movement over a 60 s period. Subsequent frames were superimposed from decompiled videos using Photoshop CS4 to calculate number of motile animals and their gliding velocity. Ciliary beating was visualized by placing live animals on coverslips with Vaseline/parafilm spacers using an Olympus BX51 microscope with DIC optics. Video segments captured using a high frame rate camera (X-PRI F1) were analyzed

using Virtualdub and ImageJ software. Detailed procedures for these assays have been described previously (*Rompolas et al., 2009*, *2013*).

## Lysate preparation and PAM enzyme assays

*C. reinhardtii* cells were collected by centrifugation at 1000 ×g for 5 min and resuspended in 20 mM NaTES, 0.2 M NaCl, 10 mM mannitol, pH 7.4, 1% Triton X-100, Roche protease inhibitor cocktail and PMSF. Cells were homogenized by two rounds of freeze thaw and sonication. Soluble fraction was collected by centrifugation at 9500 × g for 2 min at 4°C. Protein concentration was estimated using the chlorophyll method as described in (*Strenkert et al., 2016*). Immunoblot analysis revealed that this procedure solubilizes essentially all of the PAM protein in these cell wall-less strains.

Whole body planarian lysates for enzyme assays were prepared by homogenizing animals in 20 mM NaTES, 10 mM mannitol, pH 7.4, 1% Triton X-100 buffer supplemented with a protease inhibitor cocktail and PMSF. Lysates were subjected to two rounds of freeze-thaw, sonicated and tumbled at 4°C for 20 min. Soluble fractions were collected by centrifuging the samples at 17,000 ×g at 4°C for 20 min. BCA protein estimation reagents (Thermo Fisher Scientific, Waltham, MA) were used for determining protein concentration. PHM and PAL enzyme assays were carried out as previously described (*Kolhekar et al., 1997*). All enzyme assays were performed in triplicate and mean ± SD were plotted.

## Antibodies

The antibodies used are listed in *Supplementary file 2*. Antibody to the luminal enzymatic domains of CrPAM lacking the transmembrane and cytosolic domains was generated by constructing a pCIS vector (*Vishwanatha et al., 2014*) encoding residues 1–743 of CrPAM (KT033716) followed by a COOH-terminal rhodopsin epitope tag (CrPHM-PAL-rhod). The full length CrPAM construct generated previously (*Kumar et al., 2016b*) was used as a template for the PCR reaction. The pCIS-CrPHM-PAL-rhod vector was expressed in CHO-DG44 cells and stable cell lines secreting CrPHM-PAL-rhod were selected by screening spent medium for a rhodopsin-tagged ~80 kDa protein (*Vishwanatha et al., 2014*). Proteins in spent serum-free medium were concentrated ~10 fold using a Vivaflow 50 or 200 crossflow cassette with a 30 kDa molecular weight cutoff. Proteins precipitated by 60% ammonium sulfate were then dialyzed into 20 mM NaTES, pH 7.0 and CrPHM-PAL-rhod was purified by ion exchange chromatography on a 35 ml Q-Sepharose column equilibrated with 20 mM NaTES, pH 6.5 and eluted with a gradient to 20 mM NaTES, 1 M NaCl, pH 6.5 (*Figure 2—figure supplement 1*). Three pre-screened rabbits (Covance; #314, 317, 319) received primary injections of 0.2 mg CrPHM-PAL-rhod followed by booster injections of 0.1 mg CrPHM-PAL-rhod. For affinity-purification, 1 mg CrPHM-PAL-rhod was covalently linked to 1.5 ml NHS-Activated Agarose beads (Thermo Scientific). After dialysis of immunoglobulins enriched by precipitation with 45% ammonium sulfate, CrPHM-PAL-rhod antibodies bound to the affinity-resin were eluted with 0.2 M glycine, 0.1 M NaCl, 0.1% TX-100, pH 2.3 and dialyzed into 100 mM Na phosphate, pH 7.4.

## Immunofluorescence

Immunostaining of *C. reinhardtii* cells with CrPAM antibody was performed as described in (*Kumar et al., 2016b*). For CEP290 and IFT antibodies, cells were allowed to adhere on 0.1% polyethylenimine-coated coverslips and fixed for 5 min with methanol at −20°C. Subsequent blocking and antibody incubation steps were done as described in (*Craige et al., 2010*). All images were obtained with a Zeiss Axiovert 200M microscope with 63× and 100× oil objectives and AxioVision software. Maximum intensity projections of optical sections collected with the ApoTome module are shown.

## Immunoblotting and quantification

Lysates (equal protein) were separated on 4–16% or 10–20% SDS-PAGE gels (BioRad) and transferred to PVDF membranes using standard methods. Samples were heated in Laemmli sample buffer at 55°C for five minutes prior to loading on gels. Immunoblots were visualized using the Syngene Pxi imaging system and band intensities were quantified using GeneTools software.

## Scanning electron microscopy

*C. reinhardtii* samples - Cells were fixed in solution by adding equal volume of 5% glutaraldehyde (Electron Microscopy Sciences) in TAP medium for 15 min. Cells were collected by centrifugation and placed onto 0.1% polyethylenimine coated coverslips for 5 min. After removing non-adherent cells, the coverslips were incubated in 2.5% glutaraldehyde in 0.1M sodium cacaodylate, pH 7.2 for 45 min. Samples were dried in an Autosamdri-815 critical point dryer (Tousimis Research) and sputter coated before imaging in a JEOL JSM-5900LV scanning electron microscope. Cell sizes were quantified by manually tracing the outline of cells in Metamorph.

Mice – Tail clips taken from E12.5 embryos obtained after mating $Pam^{+/-}$ mice were used to determine the genotype; embryos were fixed in 2% glutaraldehyde in 1x PBS and tail clips were used to determine the genotype. Embryos were cut in half at the midline using a sharp scalpel in fixative and washed with 0.1M cadodylate buffer. Subsequent steps were performed essentially as described above for *C. reinhardtii* cells. Two pairs of WT and $Pam^{-/-}$ embryos were examined; ciliary density and length were manually measured for both sets using Metamorph. Representative plots from one of two experiments that gave similar results are shown.

Planarian samples - Animals were processed for SEM as described in (*Rompolas et al., 2010*).

## Transmission electron microscopy

*C. reinhardtii* cells - Cells were fixed in suspension by mixing with an equal volume of 5% glutaraldehyde (Electron Microscopy Sciences) in TAP medium for 15 min. Cells were pelleted and resuspended in 2.5% glutaraldehyde in 0.1M sodium cacaodylate, pH 7.2 for 45 min. Fixed cells were washed in 0.1M sodium cacodylate buffer, osmicated with osmium ferricyanide, dehydrated through an ethanol series and embedded in Epon resin. Thin sections were post-stained with uranyl acetate and lead citrate and visualized in a Hitachi H-7650 transmission electron microscope at 80kV. Two sets of control and PAM amiRNA cells were examined; images were quantified manually using Metamorph. For measurement of Golgi curvature, lines were drawn from the edge of the middle stack to the center of the Golgi and the angle at the point of intersection was measured.

Planaria - Animals were processed for TEM as described in (*Rompolas et al., 2010*).

Mice - Tail clips were taken from E12.5 embryos obtained after mating $Pam^{+/-}$ mice and used to determine the genotype; embryos were fixed in 1% glutaraldehyde and 2% paraformaldehyde in 1x PBS overnight. The fixative was replaced with 1% glutaraldehyde in 0.1 M cacodylate buffer. Subsequent steps were performed essentially as described above for TEM analysis of *C. reinhardtii* cells. Two pairs of WT and $Pam^{-/-}$ embryos were examined for primary ciliary length measurements.

## Starch measurements

Cells were concentrated by centrifugation of 15 ml of culture at 3,600 × g for 10 min, frozen in liquid nitrogen and stored at −80°C until further use. Starch was obtained from the cell pellet by ethanolic extraction as described in (*Garz et al., 2012*) with minor modifications. In brief, each frozen cell pellet was resuspended twice in 80% ethanol and heated to 95°C for 30 mins. After an additional washing step with 50% ethanol and 30 min at 95°C, starch pellets were incubated in 0.1M NaOH at 95°C for 30 min. Hydrolysis of starch was performed by adding HCl 0.5 M + acetate/NaOH 0.1 M pH 4.9 buffer. Cellular glucose levels contained in starch were determined using amyloglucosidase digestion and the Sigma glucose (HK) assay kit (Sigma-Aldrich, St. Louis, MO) according to the manufacturer's instructions.

## Secretion experiments

*Chlamydomonas* cells were cultured under phosphate deficiency (*Quisel et al., 1996*) or sulfate deficiency (*de Hostos et al., 1988*) conditions for 24 hr; spent medium was centrifuged to remove cells and debris, followed by filtration using a 0.22 µm filter. Colorimetric substrates, *p*-nitrophenyl phosphate (Thermo Fisher Scientific, Waltham, MA) and *p*-nitrophenyl sulfate (Acros Organics, New Jersey, USA) were used as described to determine alkaline phosphatase and arylsulfatase activity, respectively, in cell homogenates and spent medium (*Quisel et al., 1996*; *de Hostos et al., 1988*). The molar extinction coefficient of *p*-nitrophenol at 410 nm in alkaline pH (ε = 18,300) was used to calculate alkaline phosphatase and arylsulfatase activities.

*Chlamydomonas* cells were cultured under iron deficiency conditions for 5 days as described previously (*Allen et al., 2007*; *Glaesener et al., 2013*). Spent medium was collected and processed as described above. Cell pellets were harvested for protein determination and medium corresponding to equal amount of cell protein was analyzed by SDS-PAGE followed by silver staining (SilverSNAP kit; Pierce).

## RNA extraction, library preparation and RNA sequencing

$2–4 \times 10^7$ cells were collected by centrifugation for 5 min at 1400 x g, 4°C. RNA was extracted using the Trizol reagent as described previously (*Strenkert et al., 2011*). DNase treatment was performed using Turbo DNAse (Ambion), concentrating and cleaning with the Zymo Research RNA Clean and Concentrator−5 Kit according to the manufacturer's instructions. For RNA-Sequencing, RNA quality and quantity were determined using a Nanodrop 2000 instrument (Thermo Scientific) and a Bioanalyzer ChiP RNA 7500 series II (Agilent). A stranded Illumina RNA-Seq library was prepared by the UCLA Clinical Microarray Core with standard Kapa RNAseq library preparation. Library quality control was performed with a Bioanalyzer Chip DNA 1000 series II (Agilent) and the libraries were quantified using Qubit. Libraries were sequenced on a HiSeq 1000 sequencer (Illumina) and single end 50 bp sequences were generated. Sequences were then aligned to the *C. reinhardtii* genome (v5.5) with RNA star. Relative expression estimates were generated using Cuffdiff 2.0.2.

## Acknowledgements

We thank Maya Yankova (UCHC Electron Microscopy Facility), Andrew Yanik, Taylor LaRese and Ping Wang (UCHC Neuropeptide laboratory) for outstanding technical assistance. We also thank Dr. Branch Craige (UMass Medical School), Dr. Douglas Cole (Washington State University) and Dr. Dennis Diener (Yale University) for sharing their antibodies, and Dr. Crysten Blaby-Haas (Brookhaven National Laboratories) for helpful discussions. This work was supported by grants DK032949 (to BAE), GM051293 (to SMK) and GM042143 (to SSM) from the National Institutes of Health.

## Additional information

### Funding

| Funder | Grant reference number | Author |
| --- | --- | --- |
| National Institutes of Health | DK032949 | Betty A Eipper |
| National Institutes of Health | GM051293 | Stephen M King |
| National Institutes of Health | GM042143 | Sabeeha S Merchant |

The funders had no role in study design, data collection and interpretation, or the decision to submit the work for publication.

### Author contributions

DK, Conceptualization, Data curation, Formal analysis, Validation, Investigation, Visualization, Methodology, Writing—original draft, Writing—review and editing; DS, Investigation, Methodology, Writing—review and editing; RSP-K, MTL, Investigation, Methodology; SSM, Resources, Supervision, Funding acquisition, Writing—review and editing; REM, Conceptualization, Resources, Supervision, Investigation, Writing—review and editing; SMK, Conceptualization, Resources, Supervision, Funding acquisition, Investigation, Methodology, Writing—original draft, Project administration, Writing—review and editing; BAE, Conceptualization, Resources, Data curation, Supervision, Funding acquisition, Investigation, Methodology, Writing—original draft, Project administration, Writing—review and editing

### Author ORCIDs

Dhivya Kumar, http://orcid.org/0000-0002-3737-014X
Richard E Mains, http://orcid.org/0000-0003-1154-1331
Stephen M King, http://orcid.org/0000-0002-5484-5530
Betty A Eipper, http://orcid.org/0000-0003-1171-5557

## Ethics

Animal experimentation: All procedures involving mice were approved by the UCHC Institutional Animal Care and Use Committee (protocol 101529-1119), in accordance with National Institutes of Health and ARRIVE guidelines (https://www. nc3rs.org.uk/arrive-guidelines).

## Additional files

### Supplementary files

• Supplementary file 1. Changes in gene expression of ciliary components in control and PAM-amiRNA cells analyzed by RNA sequencing. Transcript abundance of genes encoding intraflagellar transport (IFT), transition zone (TZ), Bardet-Biedl syndrome (BBS) and trafficking components in three control and three PAM amiRNA strains. Mean ± SD RPKM (reads per kilobase per million mapped reads) values from control and PAM amiRNA strains are tabulated.

• Supplementary file 2. Antibodies used in this study. The source of each of the antibodies used in this study and the dilutions employed for immunofluorescence (IF) and western blot (WB) analysis are tabulated.

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
