## [Decision Letter]

Thank you for submitting your article "A bioactive peptide amidating enzyme is required for ciliogenesis" for consideration by *eLife*. Your article has been reviewed by three peer reviewers, and the evaluation has been overseen by Anna Akhmanova as the Senior Editor and Reviewing Editor. The reviewers have opted to remain anonymous.

The reviewers have discussed the reviews with one another and the Reviewing Editor has drafted this decision to help you prepare a revised submission.

Summary:

This manuscript describes for the first time an evolutionarily conserved role of the peptide amidating enzyme, peptidylglycine α-amidating monooxygenase (PAM), in cilium biogenesis. Using RNAi-mediated knockdown strategies (*Chlamydomonas* and planarian) or PAM knockout animals (mouse), the authors show by fluorescence- and electron microscopy that loss of PAM leads to failure of ciliary assembly beyond the transition zone (TZ) in *Chlamydomonas* and planarians, whereas a milder, yet still severe, ciliogenesis defect is seen in neuroepithelial cells of mutant mouse embryos. Furthermore, they provide evidence that total cellular levels of several ciliogenic factors (certain TZ and IFT proteins) are altered in *Chlamydomonas* cells lacking PAM. These cells also display measurable defects in the secretory pathway, including altered Golgi morphology, enlarged starch grains and defects in protein glycosylation. The authors therefore propose that PAM mainly affects ciliogenesis by regulating Golgi function and post-Golgi trafficking to cilia.

The large number of ciliopathies attests to the large number of gene products already known that participate in formation of a properly functioning cilium. The major strength of the manuscript is that it implicates an unexpected player in ciliogenesis. Thus, future research on identifying the molecular function of PAM in cells such as *Chlamydomonas* could uncover novel steps in ciliogenesis. However, there are a number of serious concerns about the nature and specificity of the observed defects, which would need to be addressed to make this paper acceptable for publication in *eLife*.

Essential revisions:

1) Because ciliogenesis was blocked in the *Chlamydomonas* PAM knockdown (kd) cells, the authors examined expression of several cilium-related genes and found some were more highly expressed and some were expressed at lower amounts. Similarly, because the Golgi provides membrane proteins to the cilium, Golgi function was also examined in the PAM kd cells and found to be altered. These experiments were done to support favored models of the defects caused by PAM kd. It seems that a more objective approach would be to determine the effects of PAM kd on multiple structures and processes in cells, and if only those related to Golgi and ciliogenesis were affected then one could conclude that PAM functions specifically in those two systems. For example, the mitochondria in the PAM kd cells in Figure 10 appear much larger, more heterogenous in shape, and more vacuolated than in wild type. Is this typical? Was mitochondrial function affected by PAM kd? Photosynthesis? Autophagy? In particular, the manuscript should report whether cell cycle or growth rate properties were affected by PAM kd. One potential concern is that the cells are just unhealthy and this leads to other described phenotypes. The authors should also indicate whether the cells are in continuous light or a light-dark cycle.

2) The manuscript does not provide sufficient data to determine whether it is the PAM enzymatic activity per se that is required for ciliogenesis. As the authors point out, in some systems the cytoplasmic portion of PAM provides scaffolding/trafficking activities that appear to be independent of the enzymatic properties of PAM, and thus it could be the scaffolding activity mediated by the residual part of the cytoplasmic domain rather than the enzymatic activity that is important in ciliogenesis. Based on their results in planaria that PMH kd alone influences ciliogenesis, the authors favor the idea that PAM enzymatic activity is required for ciliogenesis. This model would be substantively strengthened if the PAM kd *Chlamydomonas* cells were rescued with wild type and enzymatically inactive form of PAM. Rescue experiments are also important because without any results showing rescue of at least some of the observed phenotypes, it is unclear whether or not these changes are simply off-target effects or are specifically mediated by the PAM knock down. In this context it is somewhat disconcerting that the expression of certain BBS genes are altered in the PAM knock down cells ([Supplementary-material SD1-data]).

3) The manuscript straddles the fence on whether PAM disruption interferes with ciliogenesis directly, or whether PAM disruption alters Golgi properties and secretion and thereby only indirectly alters ciliogenesis. If the latter is correct, then the manuscript needs to do a much better job of explaining why this is such an important advance.

One suggested experiment is to assess secretion using the assays shown in Figure 10 in the insertional mutant of PAM1. Since the cyto-tail of PAM mediates Golgi localization, it is conceivable that CrPAM-∆CD fails to localize to Golgi and the CrPAM-∆CD cells are secretion defective. If this were the case, this would indicate that the ciliary defects in PAM knockdown cells are not caused by secretion defects.

4) Data analysis, presentation, quantification and interpretation:

A) The authors present a series of results indicating that PAM knockdown in *Chlamydomonas* affects the level of several proteins implicated in ciliary trafficking or function (e.g. CEP290 and NPHP4, Figure 7) and that this occurs at the posttranscriptional level. But it is not clear how a defect in PAM would cause such an effect, nor is it clear if this change in protein level is a general phenomenon seen in cells lacking cilia and/or having perturbed Golgi function. One way to test this would be to determine if perturbation of Golgi in general (e.g. Brefeldin A treatment or knock down of AP-1 or clathrin etc.) or IFT (using available IFT mutant cells) causes the same effects on tubulin post translational modification and TZ protein expression as PAM knock down. It would be good to include at least some additional data on the specificity of the observed changes.

B) The manuscript asserts that the distribution of PAM antibody staining in Figure 1 demonstrates that PAM is in the Golgi of *Chlamydomonas* without providing an independent marker for the Golgi. What is the evidence that the area indicated to be Golgi indeed is the Golgi? Also needed is a blot of whole cells documenting that the antibody stains just PAM. In this regard, an IF image of the ∆-cd mutant cells stained with the antibody that recognizes the cytoplasmic domain would be a useful negative control for the antibody.

C) The 2016 paper from this group on PAM in *Chlamydomonas* reported that PAM activity bound to the microtubules of isolated cilia. It was surprising that there was no increase in specific activity of the microtubule-associated enzyme, however, raising the question of the physiological significance of PAM in cilia. The current manuscript needs to present separate blots with equal protein and equal cell number equivalents to quantify the amount of PAM in cilia versus cell bodies in *Chlamydomonas*.

D) More quantitative data are needed for planaria experiments: Need to quantify percent of animals that were immotile and the frequency of reorientation of gliding direction. Did the cytoplasmic axonemes in the planaria cd cells have a basal body? Without providing the total numbers of animals counted for each sample that showed cytoplasmic axonemes, the reader cannot assess the importance of the result that 57 of the kd animals had cytoplasmic axonemes compared to 0 of the control animals.

E) Please provide quantitation to support the assertion in the subsection “PAM is required for primary cilia assembly in the developing neuroepithelium of mammals”, that ciliary density was unaffected in the pam^-/-^ neuroepithelial cells.

F) Figure 6: Authors need to explain how it is possible to use TEM to quantify ciliary length. The SEM shows that most of the cilia are bent. Did the authors use serial sectioning for the TEM to identify the ciliary tip, which would be required to determine length? If these data are not reliable, the authors might consider removing them and using another way to quantify ciliary length.

G) The authors need to indicate how the differences in amounts of unidentified proteins detected in the Con A blots support their assertion that protein glycosylation was affected by PAM kd. As best I can tell, the Con A blots are silent about glycosylation per se; they show only that some unidentified proteins that happen to bind Con A are reduced in amount. Unless the authors can provide good arguments that this analysis indeed unambiguously reports on protein glycosylation, the data may better be removed.

[Editors' note: further revisions were requested prior to acceptance, as described below.]

Thank you for resubmitting your work entitled "A bioactive peptide amidating enzyme is required for ciliogenesis" for further consideration at *eLife*. Your revised article has been favorably evaluated by Anna Akhmanova as the Senior Editor and Reviewing Editor, and two reviewers.

The manuscript has been improved but there are some remaining issues that need to be addressed before acceptance, as outlined below. In particular, the analysis and interpretation of the data describing the impact of chemical inhibitors on cilia growth needs to improved, and additional antibody validation is required.

1) Chemical inhibition of the rate of ciliary regrowth: The assertion that PBA and neocuproine inhibited the rate of ciliary regrowth is not supported by the evidence. The slopes of both lines in Figure 2 appear to be almost identical. The line for the control is above the line for the PBA sample because the cilia in the control at 10 minutes were slightly longer, but after that the rates of increase seem similar. Related to this, reporting that the individual points on the control and experimental samples in Figure 2 are significantly different, says only that the lengths at those times were different. To establish, as the text states, that the rates of growth were significantly different, the authors need to compare the slopes of the two lines. Another significant concern is that the X-axes for Figure I and J are not linear, this should be changed. Taken together, the data need to be completely re-assessed.

2) Interpreting inhibitor experiments: Even if re-assessment of the ciliary regrowth data shows some effect of the inhibitors, the bigger message is that in the presence of the PAM inhibitors, cells can regrow cilia at a rate not much lower than controls, and the cilia reach nearly the same length as controls. Taken together with the miRNA results, one interpretation of the inhibitor experiments is that if cells possess the proper structures (properly attached basal bodies and IFT machinery, for example), then they can synthesize, transport, and assemble all of the components required to grow a cilium in the absence of PAM activity. This appears to be an important finding that should be highlighted more fully by the authors.

3) The antibody against the PAM luminal domain is not properly characterized: No evidence is presented that the PAM luminal domain antibody indeed recognizes PAM. Reactivity with a band on immunoblots that migrates similarly to PAM is wholly insufficient to validate the antibody.

---

## [Author Response]

Essential revisions:

1) Because ciliogenesis was blocked in the Chlamydomonas PAM knockdown (kd) cells, the authors examined expression of several cilium-related genes and found some were more highly expressed and some were expressed at lower amounts. Similarly, because the Golgi provides membrane proteins to the cilium, Golgi function was also examined in the PAM kd cells and found to be altered. These experiments were done to support favored models of the defects caused by PAM kd. It seems that a more objective approach would be to determine the effects of PAM kd on multiple structures and processes in cells, and if only those related to Golgi and ciliogenesis were affected then one could conclude that PAM functions specifically in those two systems. For example, the mitochondria in the PAM kd cells in Figure 10 appear much larger, more heterogenous in shape, and more vacuolated than in wild type. Is this typical? Was mitochondrial function affected by PAM kd? Photosynthesis? Autophagy? In particular, the manuscript should report whether cell cycle or growth rate properties were affected by PAM kd. One potential concern is that the cells are just unhealthy and this leads to other described phenotypes. The authors should also indicate whether the cells are in continuous light or a light-dark cycle.

The absence of cilia in the *Chlamydomonas* knockdown strains was a total surprise and we now try to convey the open-minded approach we took in trying to understand it. As stated in the original manuscript, we observed enlargement of starch grains in the PAM kd strains, without a concomitant increase in total starch. We evaluated growth of the PAM kd strains and saw no difference from EV controls. As requested by the reviewers, we now include growth curves for the EV and PAM kd strains in both minimal (M) and acetate-containing (TAP) media. These demonstrate the PAM kd strains grow at the same rate as controls (Figure 1—figure supplement 2; subsection “Knockdown of PAM expression disrupts ciliogenesis in *C. reinhardtii*”, last paragraph). Thus, growth under both photoautotrophic (CO_2_ as the sole carbon source assimilated by photosynthesis) and photoheterotrophic (using acetate as a carbon source) conditions is unaltered. The reviewers asked if mitochondria were larger and more heterogeneous based on the one micrograph shown. We examined numerous micrographs and found that there was no significant difference in mitochondrial organization. This clearly indicates that the PAM kd cells are generally healthy and do not exhibit major metabolic defects in chloroplast or mitochondrial function. Since membrane trafficking is essential for ciliogenesis, we examined the contractile vacuole cycle, another process that is highly dependent on membrane trafficking. We found that the PAM kd cells had a completely normal contractile vacuole cycle under standard culture conditions (Figure 1—figure supplement 2; see aforementioned paragraph). The reviewers asked if cells were grown in continuous light or a light/dark cycle – they were grown in continuous light and this is now indicated in the Methods subsection “*C. reinhardtii* Strains and Culture Conditions”.

2) The manuscript does not provide sufficient data to determine whether it is the PAM enzymatic activity per se that is required for ciliogenesis. As the authors point out, in some systems the cytoplasmic portion of PAM provides scaffolding/trafficking activities that appear to be independent of the enzymatic properties of PAM, and thus it could be the scaffolding activity mediated by the residual part of the cytoplasmic domain rather than the enzymatic activity that is important in ciliogenesis. Based on their results in planaria that PMH kd alone influences ciliogenesis, the authors favor the idea that PAM enzymatic activity is required for ciliogenesis. This model would be substantively strengthened if the PAM kd Chlamydomonas cells were rescued with wild type and enzymatically inactive form of PAM. Rescue experiments are also important because without any results showing rescue of at least some of the observed phenotypes, it is unclear whether or not these changes are simply off-target effects or are specifically mediated by the PAM knock down. In this context it is somewhat disconcerting that the expression of certain BBS genes are altered in the PAM knock down cells ([Supplementary-material SD1-data]).

To our knowledge, rescue experiments are not the normal standard used in the *Chlamydomonas* amiRNA field (e.g. Zhao et al. [2009] Plant J 58: 157-164; Molnar et al. [2009] Plant J 58: 165-174; Dymek and Smith [2012] J Cell Sci 125: 3357-3366). However, we have tried the suggested rescue experiment but have been unable to express epitope-tagged versions of PAM. Therefore, to directly address the specificity of the knockdown obtained, we used a second amiRNA (amiRNA1), which targets a different region of the PAM gene (see updated Figure 1), and found that this also lead to reduction of PAM activity and defective ciliogenesis (Figure 1—figure supplement 3 and Table 1). Thus, the observed phenotypes are not the result of off-target effects from any given amiRNA. Furthermore, we examined multiple independently-derived amiRNA strains, all of which had the same phenotype; thus, as stated in the original manuscript, the observed phenotypes were not due to plasmid insertion at any one location within the genome. To directly address the role of PAM enzyme activity in ciliogenesis, we performed additional experiments in which we used two different methods to inhibit PAM activity: 4-phenyl-3-butenoic acid, a well-characterized, mechanism-based PHM inhibitor (Bradbury et al., [1990] Eur. J. Biochem. 189, 363-368), and neocuproine, a cell-permeant copper chelator (Mendelsohn et al., [2006] Cell Metabolism 4, 155-162), as the PHM-mediated hydroxylation reaction absolutely requires this metal. We found that both treatments resulted in a decrease in the rate of ciliary reassembly in *Chlamydomonas* following deflagellation by exposure to low pH (Figure 2 and subsection “PAM enzymatic activity is important for ciliogenesis”, last paragraph). Combined with our analysis of the C-terminal truncation mutant and the observation that reducing PHM activity alone leads to ciliary loss in planaria, these new data greatly strengthen the model that PAM enzyme activity is a key ciliogenic parameter. The reviewer comments on the observed changes in expression of two BBS genes; although flagged as significant, the measured changes are actually rather minimal (mean RPKM of 9.18 (EV) vs. 8.32 (KD) for BBS2 and 5.29 (EV) vs. 5.90 (KD) for BBS9). Therefore, we have removed reference to the BBSome.

3) The manuscript straddles the fence on whether PAM disruption interferes with ciliogenesis directly, or whether PAM disruption alters Golgi properties and secretion and thereby only indirectly alters ciliogenesis. If the latter is correct, then the manuscript needs to do a much better job of explaining why this is such an important advance.

One suggested experiment is to assess secretion using the assays shown in Figure 10 in the insertional mutant of PAM1. Since the cyto-tail of PAM mediates Golgi localization, it is conceivable that CrPAM-∆CD fails to localize to Golgi and the CrPAM-∆CD cells are secretion defective. If this were the case, this would indicate that the ciliary defects in PAM knockdown cells are not caused by secretion defects.

Our new data (Figure 2) clearly support a key role for PAM enzymatic activity in ciliogenesis, and thus provide a novel requirement for amidation in ciliogenesis. The active site of PAM can accommodate a variety of substrates (e.g. Merkler et al., [2004] Biochemistry 43, 12667-12674), and it is clear that PAM can produce both amidated peptides and amidated lipids (such as oleamide, which derives from an N-oleoylglycine precursor); amidated peptides or amidated lipids could link PAM activity to its trafficking roles. Although it is known that the Golgi is important for ciliogenesis, the factors controlling Golgi-cilia trafficking remain poorly understood. As indicated in the original manuscript, we did observe altered secretion of an iron assimilation protein and alkaline phosphatase in the PAM kd cells. However, we have now found that secretion of arylsulfatase, which occurs when cells are deprived of sulfur, is not altered in the PAM kd strains (Figure 10). Thus, the lack of PAM leads to a selective defect in secretion from the Golgi. In addition, as mentioned above, the contractile vacuole cycle, which is heavily dependent on membrane trafficking, is unaffected in the PAM kd strain under normal conditions. The selective effect of PAM kd on post-Golgi trafficking is now more clearly described in the Discussion (subsection “PAM deficiency alters Golgi ultrastructure, function and post-Golgi trafficking”, first paragraph). Furthermore, the reviewers asked if PAM was still localized to the Golgi in the CrPAM-ΔCD strain; our new immunofluorescence data indicate that this is indeed the case (Figure 2—figure supplement 1).

*4) Data analysis, presentation, quantification and interpretation:*

A) The authors present a series of results indicating that PAM knockdown in Chlamydomonas affects the level of several proteins implicated in ciliary trafficking or function (e.g. CEP290 and NPHP4, Figure 7) and that this occurs at the posttranscriptional level. But it is not clear how a defect in PAM would cause such an effect, nor is it clear if this change in protein level is a general phenomenon seen in cells lacking cilia and/or having perturbed Golgi function. One way to test this would be to determine if perturbation of Golgi in general (e.g. Brefeldin A treatment or knock down of AP-1 or clathrin etc.) or IFT (using available IFT mutant cells) causes the same effects on tubulin post translational modification and TZ protein expression as PAM knock down. It would be good to include at least some additional data on the specificity of the observed changes.

As much tubulin post-translational modification occurs in cilia (Wloga et al., [2016] CSH Perspective in Biology: Cilia pp. 17-30), we expected the aciliate PAM kd strain to exhibit reduced levels of acetylated and glutamylated tubulin; we have modified the Results to make this clear (subsection “PAM deficiency affects trafficking and transition zone proteins in *C. reinhardtii*”, first paragraph). The reviewers suggested we treat cells with brefeldin A to assess whether Golgi disruption caused the same effects on tubulin post-translational modification and transition zone protein expression as observed in the PAM kd strains. We treated control ciliated cells with brefeldin for 3 hours and observed a decrease in ciliary length, as shown previously (Dentler [2013] PLoS ONE 8(1): e53366). The levels of acetylated and glutamylated tubulin were decreased in both PAM kd and brefeldin-treated cells, consistent with loss of cilia. However, enhanced levels of the transition zone protein nephrocystin 4 were not observed following brefeldin treatment indicating that they are specific to PAM deficiency (Figure 7—figure supplement 1). In addition, brefeldin treatment increased Arf1 levels; no change in Arf1 levels was observed in the PAM kd strains. These data suggest that PAM defects are specific and not merely the result of generalized Golgi dysfunction.

B) The manuscript asserts that the distribution of PAM antibody staining in Figure 1 demonstrates that PAM is in the Golgi of Chlamydomonas without providing an independent marker for the Golgi. What is the evidence that the area indicated to be Golgi indeed is the Golgi? Also needed is a blot of whole cells documenting that the antibody stains just PAM. In this regard, an IF image of the ∆-cd mutant cells stained with the antibody that recognizes the cytoplasmic domain would be a useful negative control for the antibody.

In mammalian cells, PAM is present in the Golgi and endocytic pathway. When *Chlamydomonas* PAM is expressed in mammalian cells it shows a similar localization (Kumar et al. J Cell Sci 129: 943-956 [2016]). We previously demonstrated that the peri-nuclear staining of CrPAM in wildtype *Chlamydomonas* was dispersed following treatment with brefeldin A, indicating that CrPAM was present in the Golgi; Arf1 showed a similar localization (Kumar et al. J Cell Sci 129: 943-956 [2016]). The reviewers also asked about the specificity of our PAM-CD antibody, which was characterized in our J Cell Sci paper: cell body and ciliary immunofluorescent staining were blocked by the antigenic peptide; the antibody immunoprecipitated PAM protein and activity from *Chlamydomonas* lysates; only full-length CrPAM and the processed CD fragment were detectable by immunoblotting CrPAM-expressing HEK-293 cells. As shown in Figure 1 of the original manuscript, this antibody did not stain PAM kd cells. In *Chlamydomonas* EV and PAM kd lysates, non-specific bands are detected on immunoblots in addition to full-length PAM and the CD fragment, as now shown in Figure 1—figure supplement 3. However, no immunofluorescent staining is observed in the CrPAM-ΔCD strain (these data are now presented in Figure 2—figure supplement 1). Taken together, these data localize PAM to the Golgi area in *Chlamydomonas*.

C) The 2016 paper from this group on PAM in Chlamydomonas reported that PAM activity bound to the microtubules of isolated cilia. It was surprising that there was no increase in specific activity of the microtubule-associated enzyme, however, raising the question of the physiological significance of PAM in cilia. The current manuscript needs to present separate blots with equal protein and equal cell number equivalents to quantify the amount of PAM in cilia versus cell bodies in Chlamydomonas.

We are unclear why the reviewers found it surprising that, in our previous paper in J Cell Sci, we observed no increase in PAM specific activity in cilia compared to PAM in the cell body; these measurements were repeated multiple times. How much PAM is in cilia depends on many factors including the rate of trafficking through the secretory pathway, the rate of PAM entry into cilia, and its rate of loss from cilia, which might include recycling to the cell body and/or shedding in ciliary ectosomes. As requested, we quantified PHM specific activity in cilia and cell bodies and the% of the total PHM activity present in cilia *vs.* cell bodies; cilia contained ~7% of the total PHM activity (Figure 1—figure supplement 1), confirming our previous result demonstrating similar PHM specific activities in cilia and cell bodies. The ciliary membrane accounts for only a small fraction of the total cell membrane (Nachury, Phil Trans Royal Soc B 369, 20130465 [2014]). Further studies are needed to determine whether the 7% of total PAM present in this compartment performs a function distinct from PAM in the cell body.

D) More quantitative data are needed for planaria experiments: Need to quantify percent of animals that were immotile and the frequency of reorientation of gliding direction. Did the cytoplasmic axonemes in the planaria cd cells have a basal body? Without providing the total numbers of animals counted for each sample that showed cytoplasmic axonemes, the reader cannot assess the importance of the result that 57 of the kd animals had cytoplasmic axonemes compared to 0 of the control animals.

We now include two tables providing this information. In Table 2, for control and all experimental groups we quantified the% immotile animals, gliding velocity (mean ± SEM) and ciliary beat frequency (which was previously included in the main text). Reference to animal reorientation has been removed as it is not a readily interpretable parameter. In Table 3, we quantitated, with respect to length of epithelium examined, the number of docked basal bodies and cytosolic axonemes identified. These TEM data were obtained from two control and two knockdown animals. We have never observed cytosolic axonemes in control animals nor in our previous published knockdowns of other ciliary genes (IFT88, WDR92, outer arm dynein IC2 and LC1 and IFT dynein WDR60; Rompolas et al., Mol Biol Cell 21, 3669-3679 [2010]; Patel-King and King, Mol Biol Cell 27, 1204-1209 [2016]; Patel-King et al., Mol Biol Cell 24, 2668-2677 [2013]). It is currently uncertain whether these cytosolic axonemes emanate from a basal body; to date, we have not observed any undocked misoriented basal bodies in cross-section that might represent axoneme templating sites. This would require complete serial section reconstruction of the planaria ventral surface which is beyond the scope of this project.

E) Please provide quantitation to support the assertion in the subsection “PAM is required for primary cilia assembly in the developing neuroepithelium of mammals”, that ciliary density was unaffected in the pam^-/-^ neuroepithelial cells.

Quantitation of ciliary density on pam^-/-^ neuroepithelia was provided in Figure 6 of the original manuscript.

F) Figure 6: Authors need to explain how it is possible to use TEM to quantify ciliary length. The SEM shows that most of the cilia are bent. Did the authors use serial sectioning for the TEM to identify the ciliary tip, which would be required to determine length? If these data are not reliable, the authors might consider removing them and using another way to quantify ciliary length.

We examined cilia by TEM in multiple adjacent sections to obtain an estimate of ciliary length. We did not observe major changes in ciliary orientation by TEM even in 2 μm long cilia. We also quantified ciliary length from SEM images; these data are now provided as Figure 6. Both TEM and SEM data sets indicated that cilia were significantly shorter in the PAM^-/-^ mice than in controls. It is perhaps noteworthy that the mean length of control cilia was estimated to be slightly longer by TEM than by SEM, whereas both methods gave essentially the same result for PAM^-/-^animals. This may reflect difficulties in accurately measuring the length of the longer bent cilia by SEM in these control animals.

G) The authors need to indicate how the differences in amounts of unidentified proteins detected in the Con A blots support their assertion that protein glycosylation was affected by PAM kd. As best I can tell, the Con A blots are silent about glycosylation per se; they show only that some unidentified proteins that happen to bind Con A are reduced in amount. Unless the authors can provide good arguments that this analysis indeed unambiguously reports on protein glycosylation, the data may better be removed.

The glycosylation data has been removed, as suggested by the reviewers, and replaced with new data indicating that secretion of arylsulfatase was unaffected in PAM amiRNA cells (new Figure 10). As alkaline phosphatase and FEA1 secretion were altered, these data indicate that the effect of PAM loss on secretion is selective and not a result of generalized Golgi dysfunction.

[Editors' note: further revisions were requested prior to acceptance, as described below.]

The manuscript has been improved but there are some remaining issues that need to be addressed before acceptance, as outlined below. In particular, the analysis and interpretation of the data describing the impact of chemical inhibitors on cilia growth needs to improved, and additional antibody validation is required.

1) Chemical inhibition of the rate of ciliary regrowth: The assertion that PBA and neocuproine inhibited the rate of ciliary regrowth is not supported by the evidence. The slopes of both lines in Figure 2 appear to be almost identical. The line for the control is above the line for the PBA sample because the cilia in the control at 10 minutes were slightly longer, but after that the rates of increase seem similar. Related to this, reporting that the individual points on the control and experimental samples in Figure 2 are significantly different, says only that the lengths at those times were different. To establish, as the text states, that the rates of growth were significantly different, the authors need to compare the slopes of the two lines. Another significant concern is that the X-axes for Figure I and J are not linear, this should be changed. Taken together, the data need to be completely re-assessed.

On re-examining the plots, we realized that the way we presented the data may have led to a misunderstanding. We reanalyzed/replotted the data and included the zero-time point data. We made the x-axis linear for both plots, as requested (see updated Figure 2). We wish to be clear that it is not that the cilia on control cells were initially longer than the cilia on inhibitor-treated cells and that this difference simply continued throughout the experiment; deflagellation of *Chlamydomonas* by pH shock leads to release of the cilia immediately distal to the transition zone (e.g. Sanders and Salisbury J Cell Biol 108, 1751-1760 [1989]). As expected, in both control and experimental treatments cilia were undetectable at the zero-time point. Thus, in the presence of the inhibitors, the absolute length of the cilia, and consequently the amount of regrowth, is significantly decreased until the latest time points. The new plots clearly indicate that it takes approximately 20-30 minutes longer for neocuproine- and PBA-treated cells to regenerate ~9.5 μm cilia. This is now stated in the text (subsection “PAM enzymatic activity is important for ciliogenesis”, last paragraph). Statistical analysis (unweighted means analysis) indicates that differences between the control and experimental curves are highly significant, with P<0.0001. This is indicated in the figure legend (Figure 2 legend). The term “rate” has been removed as the rate of cilia regrowth is known to change with ciliary length (e.g. Marshall et al. Mol Biol Cell 16, 270-278 [2005]). We now state that complete ciliary regrowth is delayed following the inhibitor treatments (subsection “PAM enzymatic activity is important for ciliogenesis”, last paragraph and subsection “Possible mechanisms for the effects of PAM on ciliary assembly”, first paragraph).

2) Interpreting inhibitor experiments: Even if re-assessment of the ciliary regrowth data shows some effect of the inhibitors, the bigger message is that in the presence of the PAM inhibitors, cells can regrow cilia at a rate not much lower than controls, and the cilia reach nearly the same length as controls. Taken together with the miRNA results, one interpretation of the inhibitor experiments is that if cells possess the proper structures (properly attached basal bodies and IFT machinery, for example), then they can synthesize, transport, and assemble all of the components required to grow a cilium in the absence of PAM activity. This appears to be an important finding that should be highlighted more fully by the authors.

The reviewer makes a good point and this is now addressed in the manuscript. Also, *Chlamydomonas* stockpiles sufficient proteins in the cytoplasm to rebuild approximately ½ length cilia without additional protein synthesis. Thus, it is also possible that these cells, which are not initially PAM-deficient, already have enough amidated product(s) available to allow for ciliogenesis to occur, albeit in a delayed fashion. We have added a comment to the manuscript (subsection “Possible mechanisms for the effects of PAM on ciliary assembly”, fourth paragraph) to highlight these points, as suggested.

3) The antibody against the PAM luminal domain is not properly characterized: No evidence is presented that the PAM luminal domain antibody indeed recognizes PAM. Reactivity with a band on immunoblots that migrates similarly to PAM is wholly insufficient to validate the antibody.

We expanded the description of our luminal domain antibody validation. First, it recognizes a band that co-migrates with the PAM band detected by the anti-CD antibody (Figure 1—figure supplement 3); 2) the intensity of this band is dramatically reduced in the PAM knockdown strains compared to empty vector controls (Figure 1—figure supplement 3), indicating that the detected band is indeed PAM; 3) the intensity of this band is greatly increased in the ∆-CD strain, which directly correlates with the increase in PAM enzymatic activity of this strain (Figure 2); 4) the luminal antibody shows the same immunofluorescence staining pattern in *Chlamydomonas* as does the CD antibody (Figure 1 and Figure 2—figure supplement 2). We believe these data, which are already present in the manuscript, fully validate this antibody. We provide more information on the preparation and characterization of this antibody (subsection “PAM enzymatic activity is important for ciliogenesis”, second paragraph; subsection “Antibodies” and Figure 2—figure supplement 1 legend)and include a map of the antigen and a gel showing its purity (new Figure 2—figure supplement 1).